# STATION2RADAR: QUERY-CONDITIONED GAUSSIAN SPLATTING FOR PRECIPITATION FIELD

**Doyi Kim, Minseok Seo, Changick Kim**[*]
Korea Advanced Institute of Science and Technology (KAIST)
{doyi.kim, minseok.seo, changick}@kaist.ac.kr

## ABSTRACT

Precipitation forecasting relies on heterogeneous data. Weather radar is accurate, but coverage is geographically limited and costly to maintain. Weather stations provide accurate but sparse point measurements, while satellites offer dense, high-resolution coverage without direct rainfall retrieval. To overcome these limitations, we propose Query-Conditioned Gaussian Splatting (QCGS), the first framework to fuse automatic weather station (AWS) observations with satellite imagery for generating precipitation fields. Unlike conventional 2D Gaussian splatting, which renders the entire image plane, QCGS selectively renders only queried precipitation regions, avoiding unnecessary computation in non-precipitating areas while preserving sharp precipitation structures. The framework combines a radar point proposal network that identifies rainfall-support locations with an implicit neural representation (INR) network that predicts Gaussian parameters for each point. QCGS enables efficient, resolution-flexible precipitation field generation in real time. Through extensive evaluation with benchmark precipitation products, QCGS demonstrates over 50% improvement in RMSE compared to conventional gridded precipitation products, and consistently maintains high performance across multiple spatiotemporal scales.

## 1 INTRODUCTION

Recent data-driven models, including transformer-based (Pathak et al. (2022); Bi et al. (2023); Lam et al. (2023); Nguyen et al. (2023); Chen et al. (2023a;b) and diffusion-based (Price et al. (2025)) forecasters trained on ERA5, now rival or surpass traditional numerical weather prediction (NWP) models at medium ranges (up to 15 days).

Yet precipitation remains particularly challenging (Bonavita, 2024; Liu et al., 2024; An et al., 2025). Both operational NWP systems and current global data-driven models operate at coarse resolutions of tens of kilometers (e.g., ERA5), whereas the precipitation features most relevant for local impacts emerge at sub-grid scales, intermittently and locally.[1] This scale mismatch complicates observation and limits the usefulness of forecasts for downstream decisions. Historically, short-range precipitation prediction relied on radar echo extrapolation at its native resolution, since NWP could not resolve small-scale convection. Operational systems therefore propagate reflectivity fields using optical-flow methods such as Lucas–Kanade (Pulkkinen et al., 2019), with forecast skill fundamentally constrained by radar fidelity. Deep learning reinforced this paradigm. Radar-centric benchmarks (Veillette et al., 2020) enabled a progression of models from ConvLSTM (Shi et al., 2015) to diffusion-based nowcasting approaches (Gao et al., 2023; Yu et al., 2024a; Gong et al., 2024a;b) to achieve strong short-lead performance. However, precipitation forecasting is far from solved. Most pipelines assume radar as the primary input, but radar networks are costly and geographically limited, making these approaches feasible mainly in regions like Europe and the United States. Moreover, radar resolution is effectively fixed, limiting the representation of processes occurring below that scale.

These limitations motivate approaches that move beyond radar-only inputs. Conventional attempts to construct precipitation fields without radar have relied on statistical interpolation from gauges.

---

[*]Corresponding author.
[1]Rainfall often forms in localized, rapidly evolving structures smaller than the pixels of global models, leaving these subgrid-scale processes unresolved in numerical weather prediction models.

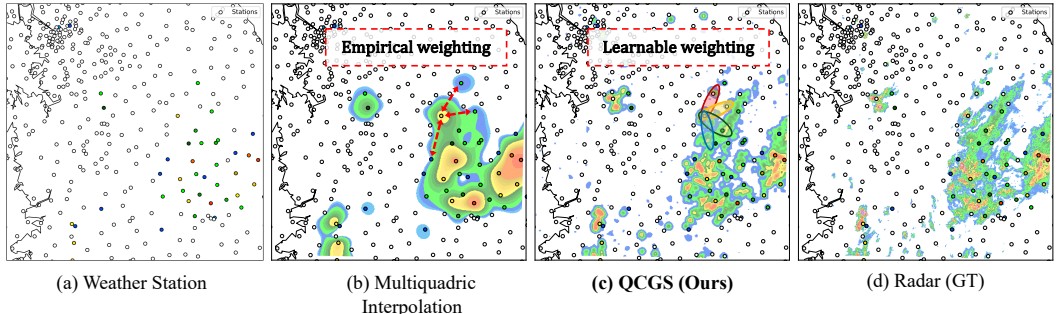

Figure 1: Construction of precipitation fields from (a) sparse AWS observations. (b) Empirical kernel-based interpolation oversmooths and blurs rainfall boundaries. (c) QCGS leverages satellite context and gauge anchors to selectively place Gaussians, producing resolution-flexible and structurally consistent fields. (d) Ground-truth radar at 2 km resolution for reference.

Methods such as Barnes interpolation, kriging, or optimal interpolation (Barnes, 1964; Alaka & Elvander, 1972; Biau et al., 1999) represent observations by assigning Gaussian weights across a grid. While effective in principle, these methods tend to blur sharp precipitation boundaries and are highly sensitive to station density and empirically chosen kernel parameters.

Recently, satellite-only approaches such as Sat2Radar (Veillette et al., 2020; Park et al., 2025) have been proposed to approximate precipitation fields directly from spaceborne imagery. However, satellite estimates carry substantial bias and uncertainty, often produce outputs only at fixed resolutions, and cannot directly leverage the physical accuracy of gauges. In parallel, several fusion-based approaches have been explored, correcting satellite products or radar imagery with gauge observations (Benoit, 2021; Ruan et al., 2025; Curcio et al., 2025). These methods improve gridded precipitation estimates by bringing them closer to ground values, but they operate strictly on fixed-resolution grids and do not cover continuous, resolution-free field reconstructions. Moreover, approaches such as Benoit (2021) require radar reflectivity as input, whereas our goal is explicitly radar-free precipitation generation.

In summary, radar-based methods have inherently limited spatial coverage and fixed resolution, making them unable to resolve fine-scale rainfall features of operational importance.

In this work, we propose combining satellite imagery with automatic weather station (AWS) measurements to generate precipitation fields without requiring radar. Our key insight is that Gaussian weighting, long used in objective analysis, is representationally equivalent to Gaussian Splatting (Kerbl et al., 2023). Traditional interpolation computes a weighted sum of point observations using Gaussian kernels. GS generalizes this idea by modeling each observation as a "Gaussian blob" with learnable parameters, enabling resolution-agnostic rendering and selective allocation of computation.

Formally, Gaussian-weighted interpolation at query location $\mathbf{x}$ is

$$f_{\mathrm{GW}}(\mathbf{x}) = \frac{\sum_{i=1}^{N} K_\sigma(\mathbf{x} - \mu_i)\, y_i}{\sum_{j=1}^{N} K_\sigma(\mathbf{x} - \mu_j)}, \tag{1}$$

where $y_i$ is the observation at station $\mu_i$ and $K_\sigma$ is a Gaussian kernel. Gaussian Splatting (GS) instead defines

$$f_{\mathrm{GS}}(\mathbf{x}) = \sum_{i=1}^{N} a_i\, K_{\Sigma_i}(\mathbf{x} - \mu_i), \tag{2}$$

with learnable amplitude $a_i$ and covariance $\Sigma_i$. Classical Gaussian weighting is recovered as a special case of GS with fixed isotropic kernels, while GS further allows anisotropy, adaptive amplitudes, and resolution-free rendering, which are key advantages for representing sharp and localized precipitation fields.

We introduce **Query-Conditioned Gaussian Splatting (QCGS)** for precipitation field generation. QCGS takes satellite imagery and automatic weather station (AWS) observations as inputs and outputs a continuous precipitation field at arbitrary resolution, without requiring radar. Unlike

standard GS, which directly fits Gaussian primitives to ground-truth fields, QCGS conditions its Gaussian parameters on satellite–AWS context, enabling generalization across regions and seasons.

QCGS consists of three components: (1) *Selective rendering*, which evaluates only precipitation-supporting regions, suppressing non-rain areas and improving efficiency. (2) *AWS–satellite fusion*, where dense satellite features provide spatial coverage and sparse AWS gauges act as accurate anchors, jointly proposing candidate Gaussian locations. (3) *INR-based parameterization*, in which an implicit neural network maps local satellite features and query locations to Gaussian parameters (amplitude and covariance), enabling adaptive, anisotropic blob shapes and resolution-free rendering.

Through this design, QCGS moves beyond traditional empirical weighting (Fig. 1) and produces high-resolution precipitation fields that preserve sharp structures, are computationally efficient, and generalize effectively.

## 2 RELATED WORK

We discuss three related areas: Gaussian Splatting for efficient field generation, Implicit Neural Representations (INR) for coordinate-conditioned modeling, and data-driven methods in meteorology.

### 2.1 GAUSSIAN SPLATTING

3D Gaussian Splatting (3DGS) (Kerbl et al., 2023) accelerates NeRF (Mildenhall et al., 2021) by representing scenes with Gaussian primitives and avoiding redundant rendering, enabling real-time performance (Wu et al., 2024; Huang et al., 2024; Yu et al., 2024b; Guédon & Lepetit, 2024; Yang et al., 2024). Recent work has extended this idea to 2D images for compression and super-resolution, such as GaussianImage (Zhang et al., 2024), Image-GS (Zhang et al., 2025), and LIG (Zhu et al., 2025), which allocate Gaussians adaptively based on gradients or frequency content. Follow-ups like GaussianSR (Hu et al., 2025), ContinuousSR (Peng et al., 2025), and GSASR (Chen et al., 2025) introduced kernel banks and feed-forward prediction for scalability and generalization. While effective, these methods remain image-specific, motivating our extension to precipitation fields.

### 2.2 IMPLICIT NEURAL REPRESENTATIONS

INR (Sitzmann et al., 2020) encodes signals as continuous coordinate-to-value mappings, widely applied to 3D scene reconstruction (Mildenhall et al., 2021; Barron et al., 2021; Martin-Brualla et al., 2021; Barron et al., 2022; Müller et al., 2022), image compression, and arbitrary-scale super-resolution (Chen et al., 2021; Yang et al., 2021; Lee & Jin, 2022; Cao et al., 2023). Its strength lies in resolution-free modeling, but INRs must query all coordinates and lack explicit spatial structure, limiting efficiency. Nonetheless, their representational flexibility motivates our query-conditioned adaptation for precipitation fields.

### 2.3 APPLICATIONS OF DEEP LEARNING IN METEOROLOGY

Deep learning has transformed meteorology, especially in precipitation nowcasting and weather prediction. ConvLSTM (Shi et al., 2015) pioneered spatiotemporal forecasting, followed by GAN-based (Ravuri et al., 2021) and transformer-based (Bi et al., 2023) approaches that rival or surpass NWP models. More recent methods span precipitation forecasting (Veillette et al., 2020; Gao et al., 2022b; Yoon et al., 2023; Gao et al., 2023; Gong et al., 2024a; Yu et al., 2024a) and global atmospheric variable prediction (Bi et al., 2023; Lam et al., 2023; Xiao et al., 2023; Chen et al., 2023b; Xu et al., 2024; Kochkov et al., 2024). Despite progress, most rely on radar or reanalysis data (e.g., ERA5). Recent data assimilation methods (Xiao et al., 2023) attempt to reduce this dependency, but to our knowledge, our work is the first to directly generate precipitation initial conditions from satellite and station data.

## 3 PRELIMINARIES

We summarize the key notions from the perspective of *2D image rendering*.

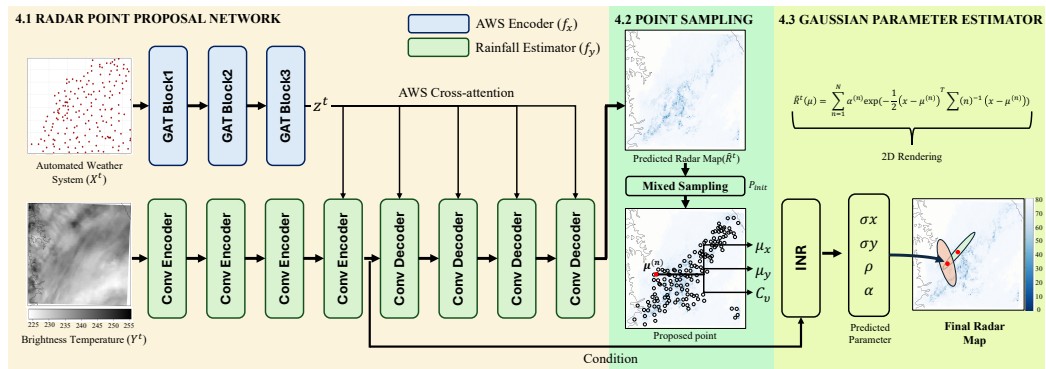

Figure 2: Overview of the proposed QCGS pipeline. AWS observations and satellite BT imagery are fused to produce a coarse surrogate field and candidate rainfall-support points. A rainfall-aware sampling strategy and an INR-based Gaussian estimator then predict splatting parameters, yielding resolution-flexible precipitation fields through selective Gaussian rendering.

**Implicit Neural Representations (INR).** An INR models an image as a continuous function

$$f_\theta : \mathbb{R}^2 \to \mathbb{R}^C,$$

that maps *spatial coordinates* $\mathbf{x} \in \mathbb{R}^2$ to pixel values. Rendering an $H \times W$ image requires evaluating $f_\theta$ at all pixel centers $\mathbf{x} \in \Omega$, which scales as $\mathcal{O}(HW)$. INRs are resolution-free and differentiable w.r.t. coordinates, but dense querying makes high-resolution synthesis slow.

**Gaussian Splatting.** 3D Gaussian Splatting (3DGS) represents a scene as a set of anisotropic 3D Gaussian primitives. Each primitive has a center $\mu_k \in \mathbb{R}^3$, covariance $\Sigma_k \in \mathbb{S}^3_{++}$, opacity $\alpha_k$, and color $c_k$. Rendering proceeds by projecting the Gaussians to the image plane, linearizing the covariance with the Jacobian of the projection, and compositing front-to-back with depth ordering:

$$I(u) = \sum_{k=1}^{K} T_k(u)\, \alpha_k\, G_k(u)\, c_k,$$

where $T_k(u)$ is the accumulated transmittance and $G_k(u)$ the screen-space Gaussian footprint.

In contrast, **2D Gaussian Splatting (2DGS)** removes geometry-specific elements and operates directly on the image plane. No 3D positions, projections, or depth ordering are required. Each primitive is simply a 2D Gaussian with center

$$\boldsymbol{\mu}_i \in \mathbb{R}^2, \qquad \Sigma_i \in \mathbb{S}^2_{++}, \qquad \alpha_i \in \mathbb{R}.$$

The rendered value at a pixel location $\mathbf{x} \in \Omega$ is

$$I(\mathbf{x}) = \sum_{i=1}^{K} \alpha_i \exp\Big( -\tfrac{1}{2}(\mathbf{x} - \boldsymbol{\mu}_i)^\top \Sigma_i^{-1} (\mathbf{x} - \boldsymbol{\mu}_i) \Big).$$

Thus, 2DGS retains the resolution-free rendering benefits of 3DGS while being simpler and computationally cheaper, making it well suited to represent sharp, localized precipitation fields.

## 4 METHOD

As illustrated in Fig. 2, QCGS follows a three-stage pipeline that fuses AWS gauge observations with satellite imagery to generate precipitation fields. Importantly, the radar point proposal network and the Gaussian rendering module are trained separately.

We first train the radar point proposal network to produce reliable rainfall-support locations, and then train the Gaussian rendering stage on top of these fixed proposals. Thus, QCGS operates as a two-stage model in terms of training, even though the full pipeline consists of three conceptual components.

## 4.1 TASK DEFINITION

We aim to estimate a high-resolution precipitation field $R^t(\mathbf{x})$ using two inputs at time $t$: a satellite image $Y^t \in \mathbb{R}^{H \times W}$ and sparse AWS measurements $X^t = \{x_i^t \mid i \in \mathcal{I}\}$ observed at irregular spatial coordinates $\{\mu_i\}$.

Formally, the goal is to learn a mapping

$$\mathcal{F}_\Theta : (Y^t, X^t) \longmapsto R^t(\mathbf{x}), \qquad \mathbf{x} \in \Omega,$$

where $\Omega$ denotes a continuous 2D spatial domain. Unlike standard Sat→Radar image-to-image translation (Park et al., 2025), our input consists of *both* a dense image and an irregular point set, and the output must be defined at *arbitrary* query locations rather than on a fixed grid.

Because the satellite image is coarse (2 km resolution) and the output precipitation field may be queried at much finer scales (e.g., 0.5 km or continuous coordinates), the task also exhibits a super-resolution nature:

$$R^t : \Omega_{\text{coarse}} \rightarrow \Omega_{\text{fine}}, \qquad |\Omega_{\text{fine}}| \gg |\Omega_{\text{coarse}}|.$$

The model parameters are estimated by minimizing reconstruction loss against radar observations during training:

$$\Theta^* = \arg\min_\Theta \mathcal{L}\big(R^t, \mathcal{F}_\Theta(Y^t, X^t)\big)$$

In summary, the task is a hybrid problem combining *image + point fusion*, *continuous field reconstruction*, and *resolution-free rendering*.

## 4.2 RADAR POINT PROPOSAL NETWORK

Automatic weather station (AWS) observations provide direct gauge measurements of rainfall. Although they offer ground truth rainfall values, the data are sparse and often contain missing values or outliers. In contrast, satellite-based brightness temperature (BT) imagery $Y^t \in \mathbb{R}^{H \times W}$ provides dense spatial coverage and is generally reliable, but it only correlates indirectly with precipitation. We combine these two complementary sources to compensate for their respective limitations.

At each time step $t \in \mathcal{T}$, the set of AWS observations is defined as

$$X^t = \{x_i^t \mid i \in \mathcal{I}\}, \quad \mathcal{I} = \{1, \dots, n\},$$

where $x_i^t$ denotes the rainfall measured at station $i$ and $n = |\mathcal{I}|$ is the number of stations. Since $X^t$ may include missing values and anomalies, we employ a graph attention network (Velickovic et al., 2017) $f_x(\cdot; \theta_x)$ to extract a robust representation:

$$z^t = f_x(X^t).$$

The satellite BT image $Y^t$ is processed by an encoder–decoder network $f_y(\cdot; \theta_y)$ to produce a dense rainfall prediction:

$$\hat{R}^t = f_y(Y^t, z^t),$$

where the AWS representation $z^t$ is fused into the decoder via cross-attention.

During training, the model parameters are optimized by minimizing the mean squared error (MSE) between the predicted rainfall $\hat{R}^t$ and the radar-derived ground truth $R^t$:

$$\mathcal{L}_{\text{MSE}} = \frac{1}{|\mathcal{T}|} \sum_{t \in \mathcal{T}} \big\| \hat{R}^t - R^t \big\|_2^2.$$

## 4.3 RAINFALL-AWARE POINT SAMPLING

In precipitation forecasting, light rain rarely leads to disasters, while heavy precipitation events are much more likely to trigger hazards and high-impact events. Therefore, the most critical objective is to accurately predict regions of heavy precipitation. Uniformly sampling points across the entire prediction field $\hat{R}^t$ is inefficient, as it treats all regions equally regardless of their importance. To

Figure 3: Visualization of different point sampling strategies for precipitation fields. (a) Ground truth radar field, (b) uniform sampling, (c) edge-based sampling, (d) heavy-rain sampling, and (e) our mixed strategy. Uniform sampling provides overall coverage but lacks details in heavy rainfall regions. Edge-based sampling emphasizes boundaries but overlooks fine-scale structure. Heavy-rain sampling concentrates points on intense precipitation, leaving light-rain areas underrepresented. In contrast, our mixed strategy balances intensity-aware sampling with spatial coverage across the field.

overcome this limitation, we design a sampling strategy that incorporates three factors: (1) gradients of $\hat{R}^t$ to emphasize edges, (2) uniform coverage within $\hat{R}^t$, and (3) rainfall intensity to prioritize heavy-rain regions.

We denote image-domain coordinates as $\mathbf{x} \in \Omega$, and write $\hat{R}^t(\mathbf{x})$ for its rainfall value.

Let $\hat{R}^t \in \mathbb{R}^{H \times W}$ be a coarse precipitation field at time $t$, and define the rain-support mask as

$$\mathcal{S}_t = \{\mathbf{x} \mid \hat{R}^t(\mathbf{x}) > \tau\},$$

with a threshold $\tau$. We then construct a convex mixture of three normalized terms:

$$P_{\text{init}}(\mathbf{x}) = \alpha \, G_{\mathcal{S}_t}(\mathbf{x}) + \beta \, U_{\mathcal{S}_t}(\mathbf{x}) + \gamma \, H(\mathbf{x}), \qquad \alpha, \beta, \gamma \geq 0, \ \ \alpha + \beta + \gamma = 1,$$

where

$$U_{\mathcal{S}_t}(\mathbf{x}) = \frac{\mathbb{1}\{\mathbf{x} \in \mathcal{S}_t\}}{\sum_{h,w} \mathbb{1}\{(h,w) \in \mathcal{S}_t\} + \varepsilon}, \quad G_{\mathcal{S}_t}(\mathbf{x}) = \frac{\mathbb{1}\{\mathbf{x} \in \mathcal{S}_t\} \, \|\nabla \hat{R}^t(\mathbf{x})\|_2}{\sum_{h,w} \mathbb{1}\{(h,w) \in \mathcal{S}_t\} \, \|\nabla \hat{R}^t(h,w)\|_2 + \varepsilon}, \quad H(\mathbf{x}) = \frac{\exp(\hat{R}^t(\mathbf{x})/T)}{\sum_{h,w} \exp(\hat{R}^t(h,w)/T)}.$$

Here, $\nabla \hat{R}^t(\mathbf{x})$ denotes the spatial gradient of the coarse precipitation field, $T > 0$ is a temperature parameter controlling the sharpness toward heavy-rain pixels, and $\varepsilon > 0$ is a small constant (set to $10^{-8}$ in our experiments) introduced to ensure numerical stability when the denominator approaches zero.

## 4.4 INR-BASED GAUSSIAN PARAMETER ESTIMATOR

Our objective is to generate dense, high-quality precipitation fields from satellite imagery and sparse AWS observations, even without radar ground truth. Conventional Gaussian splatting methods are typically optimized per image and may not generalize across scenes. We instead design an INR-based estimator that predicts Gaussian parameters only at rainfall-support queries, avoiding unnecessary computation in dry regions.

Given proposal points $\mu^{(n)} = \{(u_x^{(n)}, u_y^{(n)}, s^{(n)})\}_{n=1}^N$ from the Radar Point Proposal Network, the estimator is conditioned on intermediate satellite features $f_y(Y^t, z^t) \in \mathbb{R}^{H' \times W' \times D}$. Through cross-attention, we predict Gaussian parameters

$$\theta^{(n)} = \{\sigma_x^{(n)}, \sigma_y^{(n)}, \rho^{(n)}, \alpha^{(n)}\},$$

where $(\sigma_x^{(n)}, \sigma_y^{(n)}, \rho^{(n)})$ define the covariance $\Sigma^{(n)} \in \mathbb{S}_{++}^2$ and $\alpha^{(n)}$ controls the Gaussian amplitude. At AWS stations with nonzero rainfall, we directly set $\alpha^{(n)} = s^{(n)}$, anchoring the field to ground-truth observations.

Training minimizes reconstruction error against radar fields with regularization:

$$\mathcal{L} = \frac{1}{|\Omega|} \sum_{\mathbf{x} \in \Omega} (\tilde{R}^t(\mathbf{x}) - R^t(\mathbf{x}))^2 + \lambda_\sigma \sum_n (\sigma_x^{(n)} + \sigma_y^{(n)}) + \lambda_\alpha \sum_n \alpha^{(n)}.$$

The final precipitation map is rendered by differentiable 2D Gaussian splatting:

$$\tilde{R}^t(\mathbf{x}) = \sum_{n=1}^{N} \alpha^{(n)} \exp\Big( -\tfrac{1}{2}(\mathbf{x} - \mu^{(n)})^\top \Sigma^{(n)^{-1}} (\mathbf{x} - \mu^{(n)}) \Big),$$

with $\mu^{(n)} = (u_x^{(n)}, u_y^{(n)})$. This operator is fully differentiable, enabling end-to-end training.

## 5 EXPERIMENTS

We evaluate QCGS on satellite and AWS gauge data from 2023 and compare it against classical gridded precipitation products (IMERG from NASA, MSWEP from the University of Maryland, GSMaP from JAXA) as well as deep learning baselines based on image-to-image translation.

### 5.1 EXPERIMENTAL SETTING

**Dataset.** We use three data sources: (i) automatic weather station (AWS) gauges providing sparse point-wise rainfall measurements over land, (ii) GK2A geostationary satellite imagery (IR 10.5 μm channel, 2 km resolution), and (iii) KMA HSP weather radar fields (0.5 km resolution). We crop the study domain to a $480{\times}480$ grid (35.5°–37.8°N, 126.4°–129.1°E), where gauge density is relatively high. Models are trained on hourly data from 2019–2022 and evaluated on 2023 data. Although training is performed at 2 km resolution, we also evaluate at 0.5 km to demonstrate the ability of QCGS to render rainfall fields at arbitrary scales.

**Evaluation Metrics.** We evaluate QCGS using RMSE for overall error and LPIPS (Zhang et al., 2018) for perceptual similarity. For grid-point verification, we report Critical Success Index (CSI), Categorical CSI, Fraction Skill Score (FSS; Roberts & Lean (2008)) with a $5 \times 5$ window, and bias. We also compute Pearson and Spearman correlations to assess spatial patterns and extremes.

**Comparison Methods.** We benchmark QCGS against three categories of baselines. For classical interpolation, we use Barnes (Barnes, 1973), Multi-quadric(MQ; Nuss & Titley (1994)), and Kriging (Lucas et al., 2022) methods. For operational products, we include IMERG (Huffman et al., 2015) (NASA), a global 0.1° multi-satellite retrieval widely used in hydrology; MSWEP (Beck et al., 2019) (University of Maryland), a long-term 0.1° dataset that blends gauges, satellite, and reanalysis; and GSMaP (Mega et al., 2018) (JAXA), a near–real-time 0.1° product combining passive microwave radiometers with geostationary infrared sensors. We also include the GK2A 2-km rain rate product as a regional quantitative precipitation estimate. For data-driven baselines, we compare against NPM (Park et al., 2025), the first model to demonstrate precipitation forecasting from satellite imagery alone, where we use the satellite-to-radar stage for fairness; BBDM (Li et al., 2023), a diffusion-based image-to-image framework adapted for precipitation downscaling; and Pix2PixHD (Wang et al., 2018), a conditional GAN commonly applied to satellite-to-rainfall mapping, though it may struggle to preserve sharp convective structures.

By comparing against both operational references and learning-based models, we aim to evaluate QCGS against a broad spectrum of established standards and state-of-the-art deep methods.

**Implementation Details.** We fix the number of query points to $K{=}6000$, which provides an optimal balance between fidelity and efficiency. The surrogate radar field $\hat{R}$ is produced by a ConvNeXt-based U-Net with four encoder/decoder stages and skip connections, using Group Normalization and GELU activations. AWS observations are embedded via a three-layer Graph Attention Network (8 heads, hidden size 128) and fused with satellite features in the decoder through cross-attention.

For point selection, we adopt a rainfall-aware strategy combining edge, intensity, and uniform terms (0.3/0.4/0.3), with non-maximum suppression to avoid redundancy. Each query is passed to a five-layer MLP INR (hidden size 128 with sinusoidal positional encoding), which predicts Gaussian parameters $\{\sigma_x, \sigma_y, \rho, \alpha\}$. At AWS sites with nonzero rainfall, $\alpha$ is set directly to the observed value, anchoring the generated fields.

Training uses Adam ($1{\times}10^{-4}$ initial lr, $1{\times}10^{-5}$ weight decay, cosine schedule, gradient clipping at 1.0), with batch size 16 for 100 epochs. Regularization terms $\lambda_\sigma = 10^{-3}$ and $\lambda_\alpha = 10^{-4}$ prevent over-smoothing. All experiments are conducted on $8{\times}$NVIDIA H200 GPUs.

Table 1: Quantitative results across multiple spatiotemporal scales. Each block shows the number of evaluated cases in parentheses. QCGS is trained at 2 km and downsampled to $0.1°$ for comparison with global products. Best scores per block are in **bold**.

| Temporal scale (cases) | Category | Method | Res. | RMSE↓ | LPIPS↓ | CSI↑ | FSS↑ | CC↑ | Bias≈1 |
|---|---|---|---|---|---|---|---|---|---|
| Snapshot (1154) | Data-driven | Pix2PixHD | 0.5 km | 2.45 | 0.62 | 0.59 | 0.71 | 0.55 | 0.82 |
| | Data-driven | NPM | 0.5 km | 1.95 | 0.58 | 0.59 | 0.78 | 0.68 | 0.88 |
| | Data-driven | BBDM | 0.5 km | 1.68 | 0.54 | 0.64 | 0.84 | 0.75 | 0.93 |
| | Satellite Product | GK2A | 2.0 km | 2.89 | 0.40 | 0.20 | 0.37 | 0.12 | - |
| | Classical Interp. | Barnes | 2.0 km | 2.56 | 0.39 | 0.47 | 0.68 | 0.42 | 0.98 |
| | Classical Interp. | Kriging | 2.0 km | 2.43 | 0.40 | 0.50 | 0.69 | 0.45 | 1.03 |
| | Classical Interp. | 3DMQ | 2.0 km | 2.47 | 0.41 | 0.49 | 0.68 | 0.44 | 1.00 |
| | Ours | QCGS | 0.5 km | 1.23 | 0.49 | 0.74 | 0.91 | 0.90 | 1.02 |
| | Ours | QCGS | 2.0 km | **1.00** | **0.19** | **0.76** | **0.96** | **0.93** | 1.03 |
| Hourly mean (1154) | Satellite Product | IMERG | 0.1° | 1.66 | 0.34 | 0.50 | 0.72 | 0.42 | 0.85 |
| | Satellite Product | GSMaP | 0.1° | 1.95 | 0.38 | 0.43 | 0.64 | 0.39 | 0.78 |
| | Ours | QCGS | 0.1° | **1.33** | **0.21** | **0.66** | **0.93** | **0.74** | 0.97 |
| Daily accum. (70) | Satellite Product | IMERG | 0.1° | 14.08 | 0.33 | 0.85 | 0.92 | 0.72 | 0.95 |
| | Satellite Product | GSMaP | 0.1° | 15.89 | 0.35 | 0.92 | 0.82 | 0.70 | 0.88 |
| | Satellite Product | MSWEP | 0.1° | 12.44 | 0.32 | **0.95** | 0.91 | 0.78 | 1.07 |
| | Ours | QCGS | 0.1° | **6.68** | **0.21** | 0.93 | **0.99** | **0.95** | 1.02 |

## 5.2 QUANTITATIVE RESULTS

**Comparison with data-driven approaches.** All data-driven baselines (Pix2PixHD, NPM, BBDM) were trained and evaluated directly at 0.5 km resolution for fairness. In contrast, QCGS was trained only at 2 km resolution and later rendered to 0.5 km during evaluation. Despite this apparent disadvantage, QCGS consistently achieved the best performance across the evaluated metrics (Table 1). This robustness can be explained by two key design choices.

First, QCGS explicitly fuses AWS observations. Although gauges are sparse, their ground-level accuracy provides strong local anchors that substantially enhance field reconstruction and help correct biases that purely satellite-driven models may struggle to address. Our ablation study (Sec. 5.4) confirms this, since removing AWS information causes a sharp decline in both pixel-wise accuracy and spatial correlation.

Second, QCGS leverages Gaussian Splatting (GS) to achieve resolution-free rendering. While existing models are tied to the resolution of their training grid (for example, 0.5 km), GS allows QCGS to generate rainfall fields at arbitrary resolutions while focusing computation on rainfall-support regions. This capability preserves fine-scale convective boundaries without requiring retraining, in contrast to conventional super-resolution methods that often blur or oversmooth extremes.

Taken together, AWS fusion and GS-based resolution-free rendering explain why QCGS outperforms models trained at higher resolution. Table 1 highlights this advantage clearly, showing that even with 2 km training QCGS surpasses state-of-the-art 0.5 km baselines in both accuracy and structural fidelity.

**Comparison with classical interpolation.** Classical interpolation methods such as Barnes, Kriging, and 3DMQ rely on fixed kernel functions to spread each gauge observation across the grid. As shown in Table 1, these approaches produce smooth rainfall patterns with limited structural fidelity. Their RMSE values remain above 2.4 at 2 km resolution, and their CSI and FSS scores remain around 0.47–0.50 and 0.68–0.69, respectively. This reflects the limitation of using static, often isotropic kernels that do not explicitly adapt to precipitation geometry, leading to blurred boundaries and underestimation.

In contrast, QCGS learns anisotropic and spatially adaptive Gaussian primitives conditioned on satellite features, enabling sharper and more meteorologically consistent rainfall structures. QCGS reduces RMSE to 1.00 at 2 km, and improves CSI from 0.50 (Kriging) to 0.76 and FSS from 0.69 to 0.96, representing substantial improvements across the evaluated metrics. These results suggest that QCGS can be viewed as a learnable generalization of classical kernel-weighted interpolation, with substantially improved representational flexibility for high-resolution precipitation field reconstruction.

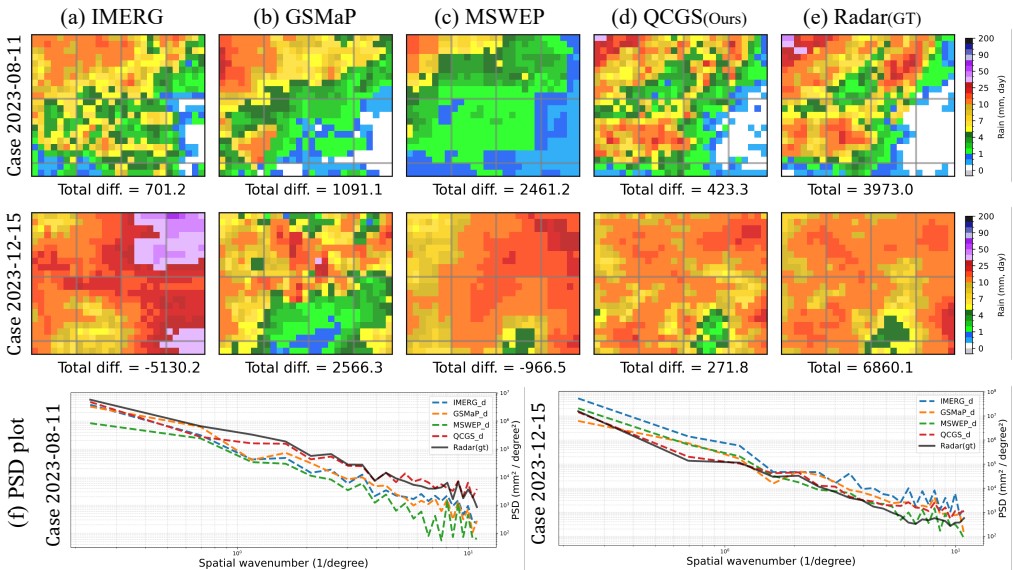

Figure 4: Panels (a)–(e) show the comparison of daily accumulated rainfall (mm, day) between radar and four rainfall products: IMERG, GSMaP, MSWEP, and QCGS. The "Total diff." values (mm, day) denote the spatially integrated difference from radar for each product. Panel (f) presents the PSD across spatial scales (since $1° \approx 111$ km, a wavenumber of 1(1/degree) corresponds approximately to a spatial scale of $\sim$110 km.)

**Comparison with operational products.** We benchmarked QCGS against operational datasets including IMERG, GSMaP, and MSWEP. These products provide global coverage, are purely satellite–driven, and apply sophisticated bias correction using rain gauges and reanalysis, which often reduces systematic errors. Nevertheless, as shown in Table 1, QCGS achieves consistently lower RMSE and higher correlation, despite being trained only with regional satellite imagery and sparse AWS measurements.

For fairness, 10-minute radar was aggregated into hourly means and daily accumulations, and all datasets were reprojected to radar coordinates using GDAL. QCGS outputs were trained at 2 km and later downsampled to $0.1°$ grids for comparison.

A key advantage of QCGS is that AWS fusion anchors local rainfall intensities, enabling sharper and more accurate regional fields than purely satellite products. At the same time, this also represents a limitation: whereas IMERG, GSMaP, and MSWEP remain purely satellite-based and thus globally deployable, QCGS currently depends on sparse but precise ground observations. In other words, QCGS delivers higher fidelity at regional scales, while operational products retain broader applicability.

## 5.3 QUALITATIVE RESULTS

**Case study and spectral analysis.** Figure 4 compares daily accumulated precipitation from radar, three operational products (IMERG, GSMaP, MSWEP), and QCGS. QCGS produces fields that are visually closer to radar, with reduced absolute differences and better preservation of localized convective cells. By contrast, GSMaP systematically underestimates intensity, while IMERG and MSWEP exhibit case-dependent over- and underestimation.

The power spectral density (PSD) analysis further shows that QCGS closely follows the radar spectrum across most scales, retaining both large-scale organization and mesoscale structure. Operational products lose variance at high wavenumbers, with MSWEP appearing oversmoothed. QCGS slightly overestimates the smallest scales, reflecting both preserved subgrid variation and minor artifacts. Overall, QCGS shows improved preservation of the spectral balance of precipitation fields relative to existing products.

Table 2: Comprehensive ablation study on CSI. (a) Effect of architecture choices: AWS fusion and Gaussian Splatting (GS) progressively improve performance. (b) Effect of sampling strategy: combining gradient, regular, and heavy-rain sampling yields the best CSI. (c) Effect of the number of query points: performance saturates around $K=6000$. Best results are in **bold**.

(a) Architecture.

| Architecture | CSI↑ |
|---|---|
| AWS (only) | 0.53 |
| U-Net (ConvNeXt) | 0.62 |
| + AWS fusion | 0.73 |
| + AWS fusion + GS (ours) | **0.76** |

(b) Sampling strategy.

| Reg. | Grad. | Heavy | CSI↑ |
|---|---|---|---|
| ✓ | | | 0.68 |
| | ✓ | | 0.71 |
| | | ✓ | 0.70 |
| ✓ | ✓ | | 0.73 |
| | ✓ | ✓ | 0.74 |
| ✓ | | ✓ | 0.72 |
| ✓ | ✓ | ✓ | **0.76** |

(c) Number of query points.

| $K$ points | CSI↑ |
|---|---|
| 1000 | 0.69 |
| 3000 | 0.72 |
| 6000 | 0.76 |
| 9000 | **0.77** |

## 5.4 Ablation Study

**Architecture (Table 2-(a)).** Starting from a U-Net (ConvNeXt) trained for satellite-to-radar translation, adding AWS fusion provides a clear improvement by anchoring rainfall intensities at gauge locations. Incorporating Gaussian Splatting (GS) further improves performance by enabling resolution-free rendering of localized rainfall, achieving the highest CSI.

**Sampling strategy (Table 2-(b)).** Regular interval sampling alone performs the worst, while gradient- or heavy-rain–based strategies provide moderate gains. Combining all three (gradient, regular, heavy-rain) yields the best CSI (0.76), highlighting the importance of jointly covering boundaries, background regions, and rainfall extremes.

**Number of query points (Table 2-(c)).** Increasing the number of sampled points $K$ steadily improves CSI up to $K=6000$, where the score reaches 0.76. Using more points ($K=9000$) yields only a marginal gain (0.77) while increasing computation, so we adopt $K=6000$ as the default trade-off between accuracy and efficiency.

## 6 Conclusion

We introduced Query-Conditioned Gaussian Splatting (QCGS), a framework for generating high-quality precipitation fields from sparse and heterogeneous observations. By treating each observation as a Gaussian kernel and conditioning splatting on satellite imagery, QCGS selectively renders rainfall regions, reducing computation while preserving sharp boundaries. The integration of Implicit Neural Representations further enables resolution-free parameterization and improved generalization across regions and seasons. Extensive experiments indicate that QCGS reduces representativeness errors, reconstructs rainfall even in gauge-sparse settings, and produces resolution-flexible fields that align closely with radar observations. These outputs are valuable not only for data assimilation but also as high-quality training data for data-driven forecasting, bridging the gap between point-based and gridded products. Overall, QCGS provides a scalable and physically consistent approach to multi-source precipitation integration, offering a promising pathway for enhancing both traditional NWP systems and emerging AI-based weather prediction models.

**Limitations and Future Work** Despite its strengths, QCGS has two main limitations. First, the method relies on automatic weather station (AWS) data to anchor rainfall intensities. In regions with insufficient gauge networks, its applicability is therefore limited. Second, our experiments were confined to the regional scale; scaling up the approach to the global domain remains an open challenge.

Looking forward, we see two promising directions. An intriguing observation is that QCGS-generated fields already align more closely with AWS measurements than raw radar reflectivity, even without any reflectivity-to-rainfall correction (Fig. 6). This suggests that QCGS may reduce certain biases present in conventional radar-derived products. Future work will further investigate this property, with the long-term goal of extending QCGS toward a scalable, global system that can complement or even substitute radar-based precipitation monitoring.

ACKNOWLEDGMENTS

This work was supported by the Institute of Information and Communications Technology Planning and Evaluation (IITP) grant funded by the Korean government (MSIT) (No. RS-2025-02263031).

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

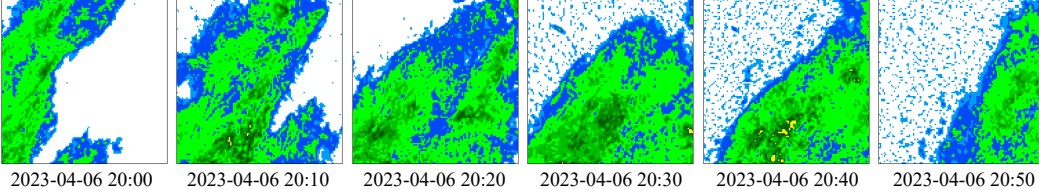

| 2023-04-06 20:00 | 2023-04-06 20:10 | 2023-04-06 20:20 | 2023-04-06 20:30 | 2023-04-06 20:40 | 2023-04-06 20:50 |

Figure 5: Examples of consecutive one-hour QCGS-generated frames. The last three frames exhibit rainfall patterns that were absent in the first three, indicating temporal inconsistencies. Such frame-to-frame mismatch can hinder the performance of video prediction models that rely on coherent temporal dynamics.

# A  APPENDIX

## A.1  INFERENCE DATA FOR PRECIPITATION FORECASTING

We further evaluated the utility of QCGS-generated radar fields as inference inputs for data-driven precipitation forecasting models. Specifically, we tested three representative baselines: MetNet-v2 (Sønderby et al., 2020), SimVP (Gao et al., 2022a), and PreDiff (Gao et al., 2023). We followed a standard nowcasting protocol in which seven past frames at ten-minute intervals are used as input and six future frames (up to +60 minutes) are predicted. MetNet-v2 directly predicts precipitation at the target lead time, while PreDiff and SimVP follow a many-to-many forecasting scheme.

All baselines were originally trained only in the radar to radar setting, and we performed no retraining or adaptation when using QCGS inputs. Despite this clear train to test mismatch, QCGS-driven forecasting still preserved meaningful predictive skill. As summarized in Table 3, the CSI at the 1 mm threshold decreased from 0.664 to 0.381 for PreDiff and from 0.591 to 0.252 for SimVP. MetNet-v2 showed only a small decrease, from 0.390 to 0.374.

We attribute this degradation to two main factors. First, QCGS does not currently enforce temporal coherence across frames, and this results in inconsistencies in the time dimension (see Fig. 5). Second, QCGS produces fields that are closer to AWS gauge values, while radar reflectivity is empirically calibrated to rain rate through the standard $Z$-$R$ relationship. This creates a mismatch for forecasting models that were trained only with radar inputs.

The smaller degradation observed in MetNet-v2 is consistent with its single-step prediction design, which is less sensitive to inter-frame consistency than many-to-many models.

Future work includes extending QCGS with temporal conditioning to provide coherent dynamics across consecutive frames, and retraining downstream forecasting models directly on QCGS-generated inputs. This may reduce the performance gap between QCGS-based and radar-based forecasting.

Table 3: Forecasting performance at +60 minutes using QCGS-generated radar fields as inputs. Baselines were trained only on radar-to-radar data and used without retraining.

| Model | CSI@1mm (R→R) | CSI@1mm (QCGS→R) |
|---|---|---|
| MetNet-v2 Sønderby et al. (2020) | 0.390 | 0.374 |
| SimVP Gao et al. (2022a) | 0.591 | 0.252 |
| PreDiff Gao et al. (2023) | 0.664 | 0.381 |

## A.2  QCGS VS. RADAR

Radar rainfall products are derived by converting reflectivity ($Z$) into rain rate ($R$) through empirical $Z$–$R$ relations. As such, they are not direct rainfall measurements and often suffer from systematic biases, especially in convective storms or orographically complex regions. In contrast, QCGS is trained on radar targets but incorporates AWS anchors at inference. Interestingly, the resulting fields

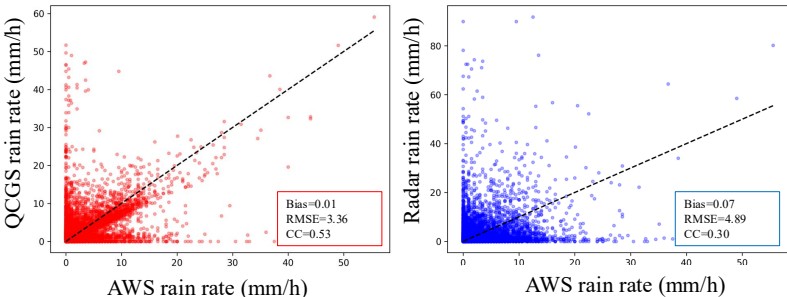

Figure 6: Bias, RMSE, and correlation coefficient (CC) of AWS rain rate compared with QCGS and radar. QCGS consistently achieves lower bias and RMSE and higher CC relative to radar, demonstrating closer agreement with gauge observations.

often align more closely with gauge observations than radar-derived rainfall. This suggests that QCGS not only reproduces radar-like spatial patterns but also implicitly corrects radar biases by leveraging point-level AWS data.

Figure 6 provides quantitative evidence: compared to radar, QCGS achieves lower bias and RMSE and higher correlation coefficients when evaluated against AWS observations. These improvements indicate that the inclusion of AWS anchors yields rainfall fields that are more consistent with ground observations.

Figure 7 presents case studies where gridded fields are directly matched with AWS locations. Here, QCGS preserves rainfall intensity more faithfully relative to gauge measurements, particularly in high-rainfall events. Importantly, AWS evaluations were performed using standard point-to-grid matching with spatial averaging, ensuring that the observed improvements are not an artifact of directly injecting AWS values but reflect genuine gains in field representation.

Taken together, these findings highlight a potential paradigm shift: QCGS offers rainfall maps that are simultaneously radar-consistent and gauge-calibrated, bridging the gap between remote sensing products and in-situ truth. In the long term, this property suggest that QCGS could serve as a complementary approach to radar-derived rainfall estimates, particularly in settings where gauge information is available.

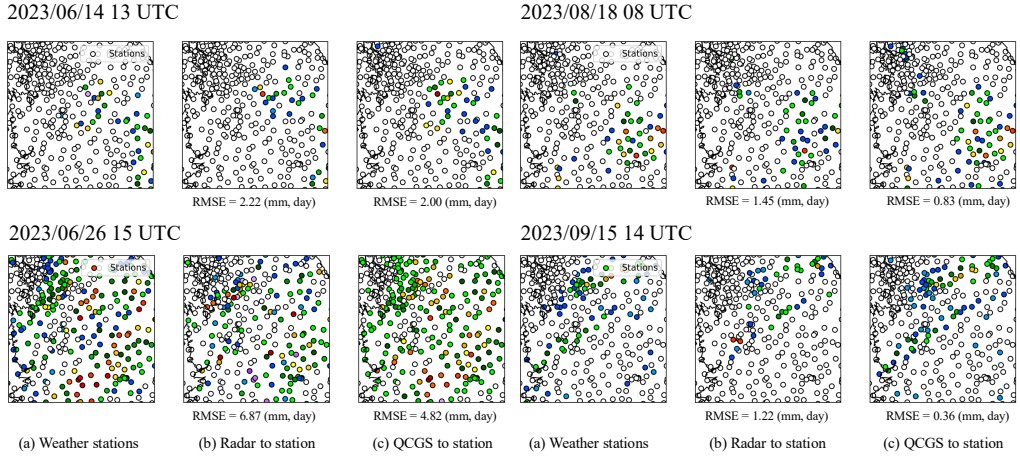

Figure 7: Case studies comparing QCGS and radar against AWS stations. Gridded fields are spatially matched to AWS locations, showing that QCGS preserves local rainfall intensities more faithfully than radar.

Table 4: CSI scores at different rainfall thresholds (mm/hour) using hourly data. QCGS is evaluated at multiple spatial resolutions (0.5, 2, and 10 km).

| Threshold | QCGS (0.5 km) | QCGS (2 km) | QCGS (10 km) | IMERG | GSMaP |
|---|---|---|---|---|---|
| 1 | 0.657 | 0.703 | 0.506 | 0.366 | 0.308 |
| 5 | 0.415 | 0.483 | 0.306 | 0.140 | 0.129 |
| 10 | 0.311 | 0.401 | 0.232 | 0.046 | 0.065 |

Table 5: Categorical POD, FAR, and CSI scores at different rainfall thresholds (mm, day) for daily accumulation data.

| | POD | | | FAR | | | CSI | | |
|---|---|---|---|---|---|---|---|---|---|
| Threshold | 10 | 50 | 100 | 10 | 50 | 100 | 10 | 50 | 100 |
| QCGS | **0.703** | **0.579** | **0.646** | **0.125** | **0.329** | **0.423** | **0.657** | **0.455** | **0.434** |
| IMERG | 0.679 | 0.369 | 0.117 | 0.267 | 0.616 | 0.614 | 0.541 | 0.173 | 0.039 |
| GSMaP | 0.591 | 0.358 | 0.277 | 0.262 | 0.710 | 0.754 | 0.493 | 0.165 | 0.119 |
| MSWEP | 0.714 | 0.315 | 0.096 | 0.286 | 0.554 | 0.584 | 0.553 | 0.191 | 0.067 |

## A.3 ADDITIONAL QUANTITATIVE ANALYSIS

Beyond continuous metrics such as RMSE and correlation, it is important to evaluate precipitation skill in a categorical manner across different rainfall intensities. To provide a more complete assessment, we present two complementary threshold-based analyses.

Table 4 reports CSI scores at 1, 5, and 10 mm using hourly data. These thresholds reflect light, moderate, and heavy rainfall. QCGS consistently outperforms satellite products across all intensity levels, and the improvement is most pronounced for heavy rainfall, where accurate detection is crucial.

To complement the hourly evaluation, Table 5 presents daily POD, FAR, and CSI metrics at 10, 50, and 100 mm per day. This daily-scale analysis captures the model's ability to detect accumulated precipitation extremes, which are critical for hydrological and disaster-related applications. QCGS achieves the best POD and CSI across all daily thresholds, while maintaining reasonable FAR values. In contrast, satellite products either miss many high-rainfall days or exhibit high false-alarm rates.

Together, the hourly and daily analyses provide a comprehensive characterization of model performance. QCGS consistently surpasses satellite products across all intensity levels and temporal scales, confirming its ability to reconstruct precipitation structure more faithfully than existing methods.

## A.4 CROSS-DOMAIN EXPERIMENTAL RESULTS

As shown in Fig. 8, we use Regions 1 and 2 (top) for training, while Regions 3 and 4 (bottom) are excluded from training. Table 6 presents the experimental results. QCGS shows only a small performance drop in unseen regions. We believe this is due to two reasons: (1) although the regions differ, they are geographically close and share similar meteorological patterns, and (2) the number of activated AWS stations varies significantly depending on the rainfall intensity. For example, heavy-rain days may activate more than 700 AWS stations, while light-rain days may activate fewer than 100. This naturally exposes the model to diverse spatial AWS configurations during training.

## A.5 VISUAL QUALITY ABLATION STUDY

Figure 9 presents a qualitative comparison among Radar, QCGS, AWS-only, and Satellite-only baselines. The AWS-only reconstruction exhibits isolated Gaussian blobs, which occur because point-based gauge measurements cannot fully represent the entire spatial domain. The Satellite-only baseline appears noticeably blurred, largely due to relying solely on pixel-wise MSE loss without ground-level anchors. In contrast, QCGS produces sharper, more coherent precipitation structures that closely resemble radar observations, benefiting from its Gaussian splatting–based rendering and

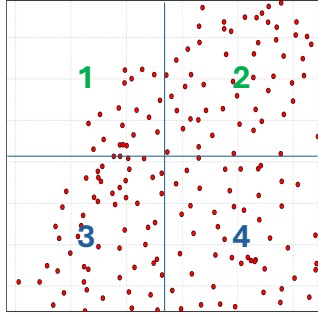

Figure 8: Training and evaluation regions. Regions 1 and 2 are used for model training, while Regions 3 and 4 are excluded.

| Metric | Cross-domain | In-domain |
|---|---|---|
| RMSE↓ | 1.01 | 1.00 |
| CSI↑ | 0.76 | 0.76 |
| Bias=1 | 1.03 | 1.03 |
| FSS=1 (ne=5) | 0.96 | 0.96 |
| LPIPS↓ | 0.25 | 0.19 |
| pCC↑ | 0.93 | 0.93 |
| rCC↑ | 0.91 | 0.92 |

Table 6: Cross-domain evaluation results.

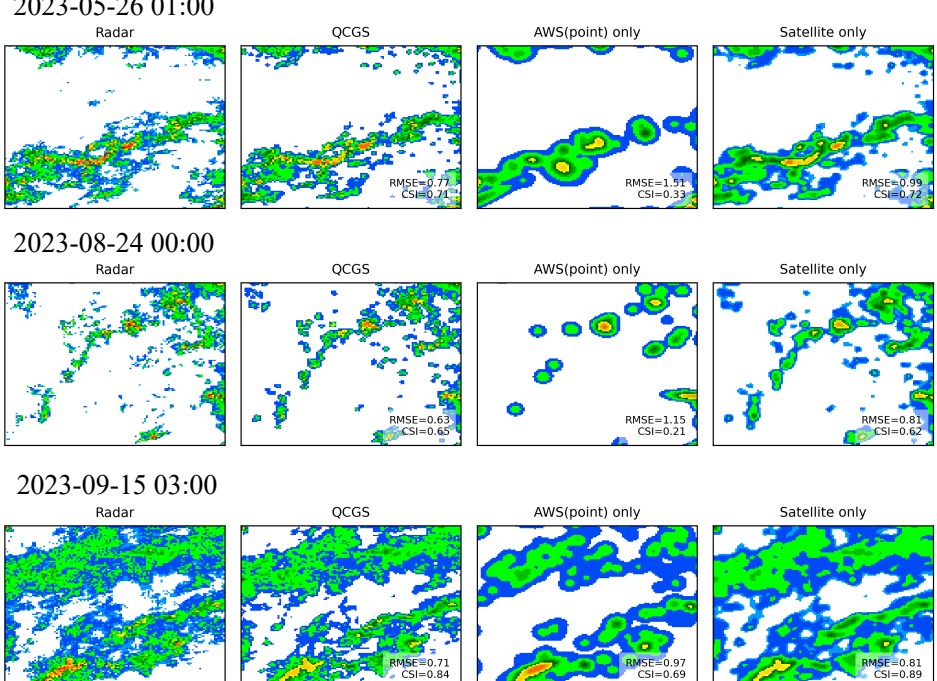

Figure 9: Visual comparison of Radar, QCGS, AWS-only, and Satellite-only across different cases.

AWS–satellite fusion. These visual results further confirm that QCGS delivers superior perceptual fidelity compared to other ablated variants.

### A.6 ADDITIONAL QUALITATIVE ANALYSIS

This section reports qualitative examples, which are randomly sampled rather than cherry-picked, to ensure fair illustration of model behavior.

Figure 10 highlights a representative case. Radar reports an area of intense rainfall, whereas QCGS produces a similar spatial pattern but with lower intensity. At first glance, this could be interpreted as an underestimation by QCGS. However, inspection of AWS gauge measurements (case: 2023-03-12 04:00) reveals that strong rainfall was not observed at ground level. This indicates that in this instance, radar likely overestimated rainfall intensity, while QCGS produced fields more consistent with in-situ truth. Such cases highlight the value of incorporating gauge anchors, which allow QCGS to mitigate biases inherent in radar-only products.

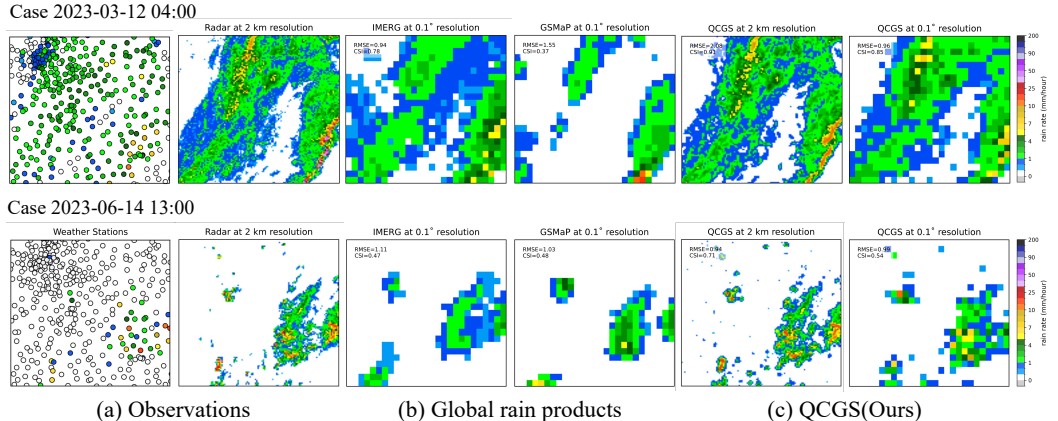

Figure 10: Qualitative comparison of observations, global rainfall products, and QCGS. QCGS preserves fine-scale precipitation structures and reduces large-scale biases relative to conventional products.

Figure 12- 13 show selected challenging cases in which QCGS underperforms relative to conventional products. While QCGS achieves competitive performance in most cases, these examples reveal limitations in capturing the spatial extent and intensity of rapidly evolving convective systems, likely due to limited constraint from sparse AWS anchors. These cases emphasize that QCGS is not universally superior under all conditions and point to potential directions for improvement, such as incorporating temporal coherence or additional observation sources.

## A.7    VISUAL QUALITY COMPARISON WITH CLASSICAL INTERPOLATION

Figure 11 presents a qualitative comparison between classical interpolation methods (Barnes, Kriging, and 3D Multiquadric) and the AWS-only variant of QCGS. All methods are evaluated under identical input, target, and output conditions to ensure a fair comparison. As shown in the figure, QCGS produces noticeably sharper and more coherent precipitation structures compared to classical approaches, demonstrating superior visual quality.

2023-05-05 16:00

2023-05-28 04:00

2023-08-30 19:00

2023-11-05 09:00

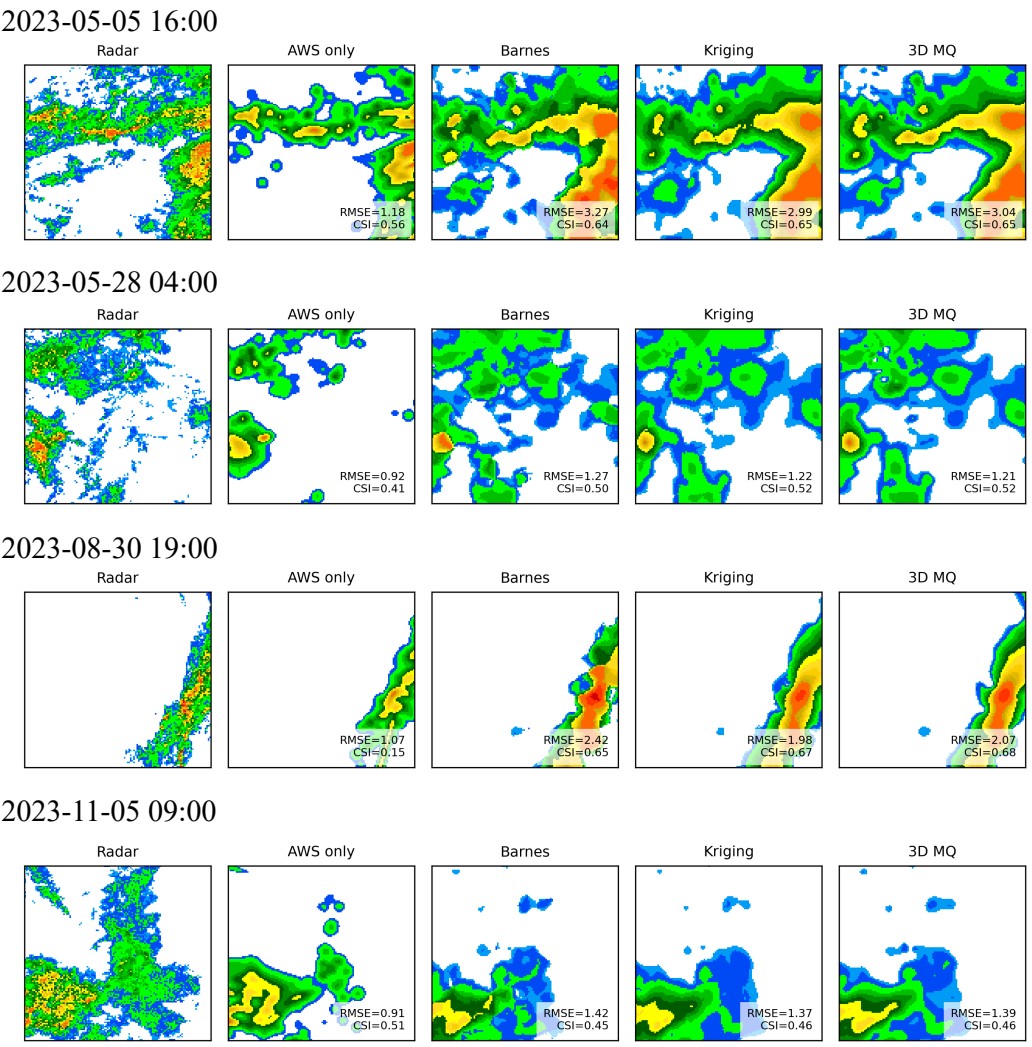

Figure 11: Qualitative comparison between classical interpolation methods and QCGS (AWS-only).

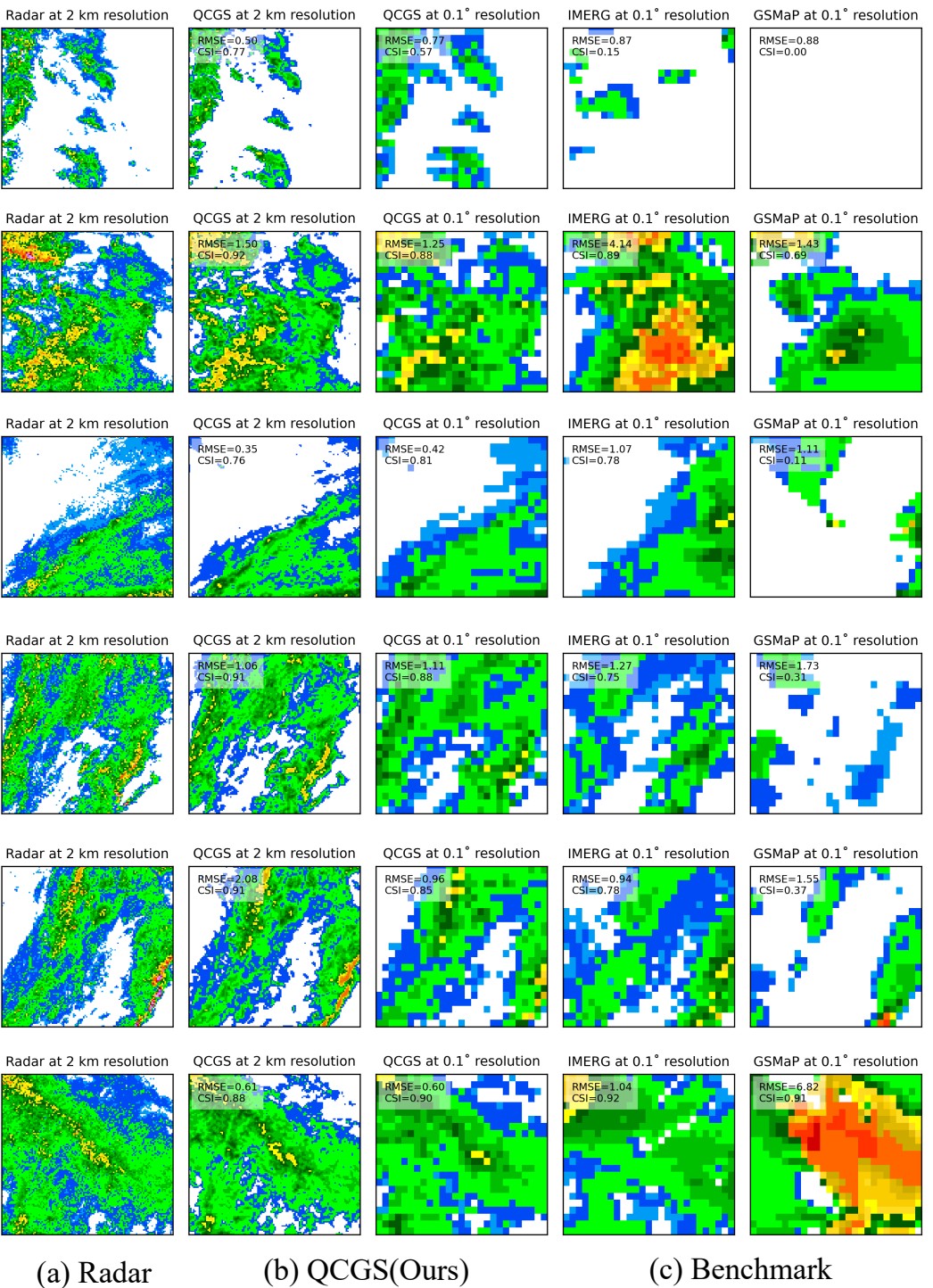

(a) Radar      (b) QCGS(Ours)      (c) Benchmark

Figure 12: Qualitative comparison of precipitation fields from radar, QCGS, IMERG, and GSMaP. Radar provides the reference, while QCGS preserves fine-scale structures more faithfully than global products. IMERG and GSMaP show smoother fields with biases in convective regions.

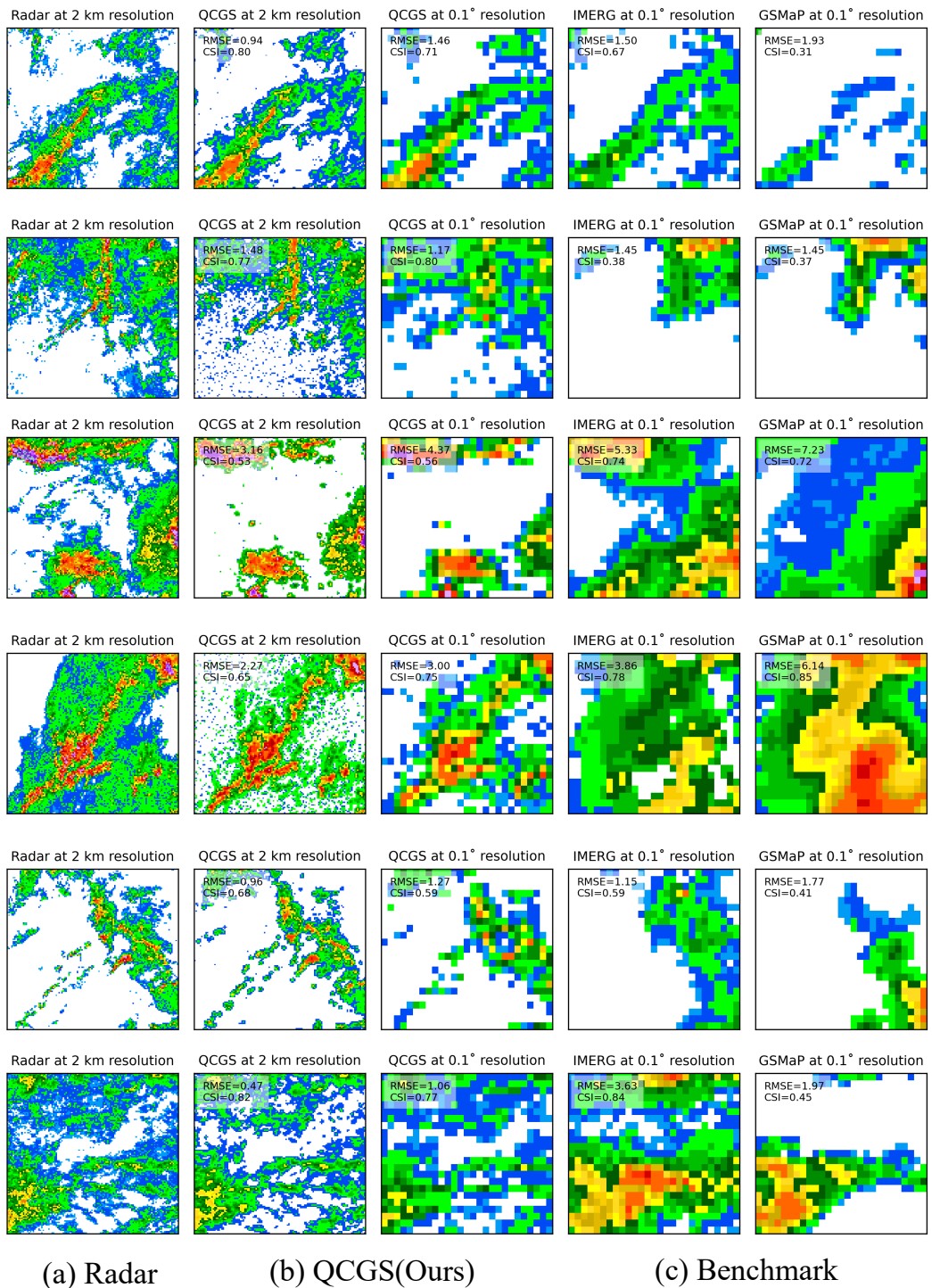

(a) Radar · (b) QCGS(Ours) · (c) Benchmark

Figure 13: Qualitative comparison of precipitation fields from radar, QCGS, IMERG, and GSMaP. Radar provides the reference, while QCGS preserves fine-scale structures more faithfully than global products. IMERG and GSMaP show smoother fields with biases in convective regions.

