# OpenReview forum: "Station2Radar: Query‑Conditioned Gaussian Splatting for Precipitation Field"
_ICLR.cc/2026/Conference — ICLR 2026 Poster_

### Official Review · Reviewer_M5eP · 2025-10-29

**Soundness:** 4
**Presentation:** 3
**Contribution:** 3
**Rating:** 4
**Confidence:** 4

**Summary:**

The paper proposes a method for fusing sparse accurate measurements to complement the dense satellite imagery. The method belongs to radar-free precipitation forecasting, motivated by the cost of radars and their limited geographic coverage. For the reasons of computational efficiency, the authors use selective rendering, where they choose to render only areas with precipitation. They fuse dense satellite data with sparse raingauge data using a customised architecture of Graph Attention for sparse raingauge observations and convolutional architecture for the dense satellite imagery, and then they perform rendering using implicit neural representation (INR) based parameterisation, mapping satellite features and locations into Gaussian parameters (amplitude and covariance), which is used in a Gaussian splatting rendering. The advantage of such rendering is that it ensures that the results are scale-invariant.

**Strengths:**

The paper tackles an important topic of fusion between the sparse raingauge data and the dense satellite imagery.

Quality: the paper is well written, and to the best of my knowledge, the methodology does not contain errors

Clarity: the paper is clearly written and seems to be reproducible

Significance: the proposed methodology is complementary to the existing ones using radar imagery

**Weaknesses:**

Originality: while the method looks to me original, there are a number of publications on similar topic, Fusion of satellite and gauge precipitation observations [1,3], as well as different but linked radar and rain gauge data fusion [2] (which is different from the topic yet it may be comparable methodologically). I would suggest the authors contrast the methodology, and importantly, problem statement to the proposed one.

Significance: while the method is sound, it is based on the successfully combined existing methods (graph attention network, convolutional model, implicit neural representations, Gaussian splatting), so it would be good it seems like the motivation hinges on the novelty of application. It is important therefore to outline explicitly, perhaps in the introduction, how the authors see the contributions

Quality: while the authors are performing the ablation study removing rain gauges, it would be also quite beneficial to see if one can execute an ablation study removing the satellite imagery (i.e. removing the convolutional features from the Radar point proposal network). I would expect the quality to drop significantly, but it would be interesting to see for the completeness of analysis.

[1] Ruan et al (2025) Fusion of satellite and gauge precipitation observations through coupling spatio-temporal properties with tree-based machine learning, Journal of Hydrology

[2] Benoit (2020) Radar and Rain Gauge Data Fusion Based on Disaggregation of Radar Imagery

[3] Curcio et al (2025) Towards a Spatiotemporal Fusion Approach to Precipitation Nowcasting, arXiv

**Questions:**

1. It would be interesting to know, given that the method also uses the dense satellite imagery, whether it is possible to use it in a radar imagery in a same way as for the satellite imagery.

The current rating reflects that there're outstanding questions related to the significance and comparison with the existing work. I would expect author to clarify upon these.

---

> ### Author Response · Authors · 2025-11-21
>
> We sincerely thank the reviewer for the detailed and highly positive assessment of our work. We greatly appreciate the recognition of our approach as a sound, well-presented, and meaningful contribution to radar-free precipitation estimation. The reviewer’s acknowledgment of the importance of fusing sparse rain gauge data with dense satellite imagery, as well as the clarity and reproducibility of our methodology, is highly encouraging to us. We have carefully addressed all reviewer comments and revised the manuscript accordingly. All changes are highlighted in magenta for clarity. **The paper has been revised and uploaded.**
>
> -----
>
> **[W1.]** Originality: while the method looks to me original, there are a number of publications on similar topic, Fusion of satellite and gauge precipitation observations [1,3], as well as different but linked radar and rain gauge data fusion [2] (which is different from the topic yet it may be comparable methodologically). I would suggest the authors contrast the methodology, and importantly, problem statement to the proposed one.
>
> **[Response]** Thank you for pointing us to the related works [1–3]. Although these studies combine satellite or radar data with gauges, their problem formulation is fundamentally different from ours. All prior works operate on fixed-resolution grids, whereas QCGS reconstructs precipitation as a continuous, resolution-free implicit field that can be rendered at any scale via Gaussian splatting. Moreover, Benoit [2] and many fusion methods assume access to radar imagery, while QCGS is explicitly designed for radar-free regions, which none of the previous approaches address. These core differences clarify that QCGS is not a variant of existing gridded fusion models but a distinct framework for continuous, any-scale precipitation reconstruction. To make this distinction clearer, we have strengthened the related work section accordingly. (L74-80)
>
> **[W2.]** Significance: while the method is sound, it is based on the successfully combined existing methods (graph attention network, convolutional model, implicit neural representations, Gaussian splatting), so it would be good it seems like the motivation hinges on the novelty of application. It is important therefore to outline explicitly, perhaps in the introduction, how the authors see the contributions
>
> **[Response]** ]First, our work is the first to combine INR and Gaussian Splatting for precipitation data, and we show that classical Gaussian interpolation and GS share the same underlying mathematical structure. This new connection enables truly resolution-free precipitation reconstruction, which existing fusion or interpolation models cannot provide. Second, directly applying GS to rainfall fields is not feasible because precipitation is highly sparse; naïve splatting introduces dense artifacts in non-rain areas. Our INR-conditioned Gaussian parameterization resolves this issue by activating Gaussians only where rainfall is physically meaningful. Therefore, the novelty of our work is not limited to the application domain. We introduce a new INR-conditioned GS framework that makes Gaussian Splatting workable for meteorological fields for the first time. We have clarified these contributions more explicitly in the introduction.
>
> **[W3.]** Quality: while the authors are performing the ablation study removing rain gauges, it would be also quite beneficial to see if one can execute an ablation study removing the satellite imagery (i.e. removing the convolutional features from the Radar point proposal network). I would expect the quality to drop significantly, but it would be interesting to see for the completeness of analysis.
>
> **[Response]** Thank you for the helpful suggestion. Following your feedback, we added the requested ablation study that removes the satellite imagery. Specifically, Table 2-(a) in the main text now includes the Satellite-removed (AWS-only) variant, and Appendix Figure 10 provides the corresponding qualitative comparisons. In addition, Appendix Figure 11 compares this AWS-only setting against classical meteorological interpolation methods under identical inputs. As expected, the AWS-only variant shows the largest performance drop due to the sparse spatial coverage of gauge stations. Nevertheless, it still outperforms classical interpolation methods, highlighting the advantage of our learned representation. These results confirm that satellite imagery is an important modality for QCGS, and they complete the ablation analysis requested by the reviewer.

---

> > ### Author Response · Authors · 2025-11-21
> >
> > **[Q1.]**  It would be interesting to know, given that the method also uses the dense satellite imagery, whether it is possible to use it in a radar imagery in a same way as for the satellite imagery.
> >
> > **[Response]** ]Yes, it is technically possible to use radar imagery in the same way as satellite imagery, since QCGS can accept any dense 2D spatial field through the convolutional encoder. However, if the goal were to reconstruct radar from radar, the task would become essentially an identity mapping problem, and the model would likely overfit to copying the input. For applications such as radar super-resolution, the framework could indeed be adapted, but it would require an additional regularization strategy to prevent trivial identity solutions. Since our goal in this work is specifically radar-free precipitation reconstruction, we intentionally do not use radar imagery as an input modality.

---

> ### Comment · Reviewer_M5eP · 2025-11-27
>
> [Thank you for pointing us to the related works [1–3]. Although these studies combine satellite or radar data with gauges, their problem formulation is fundamentally different from ours. All prior works operate on fixed-resolution grids, whereas QCGS reconstructs precipitation as a continuous, resolution-free implicit field that can be rendered at any scale via Gaussian splatting. Moreover, Benoit [2] and many fusion methods assume access to radar imagery, while QCGS is explicitly designed for radar-free regions, which none of the previous approaches address. These core differences clarify that QCGS is not a variant of existing gridded fusion models but a distinct framework for continuous, any-scale precipitation reconstruction. To make this distinction clearer, we have strengthened the related work section accordingly. (L74-80)]
>
> [First, our work is the first to combine INR and Gaussian Splatting for precipitation data, and we show that classical Gaussian interpolation and GS share the same underlying mathematical structure. This new connection enables truly resolution-free precipitation reconstruction, which existing fusion or interpolation models cannot provide. Second, directly applying GS to rainfall fields is not feasible because precipitation is highly sparse; naïve splatting introduces dense artifacts in non-rain areas. Our INR-conditioned Gaussian parameterization resolves this issue by activating Gaussians only where rainfall is physically meaningful. Therefore, the novelty of our work is not limited to the application domain. We introduce a new INR-conditioned GS framework that makes Gaussian Splatting workable for meteorological fields for the first time. We have clarified these contributions more explicitly in the introduction.]
>
> **I have checked the explanation, including in the intro and the related section, and I am happy with this answer. Thank you very much!  I think it addresses my main outstanding concerns, and I am raising the score accordingly.**
>
> [Yes, it is technically possible to use radar imagery in the same way as satellite imagery, since QCGS can accept any dense 2D spatial field through the convolutional encoder. However, if the goal were to reconstruct radar from radar, the task would become essentially an identity mapping problem, and the model would likely overfit to copying the input. For applications such as radar super-resolution, the framework could indeed be adapted, but it would require an additional regularization strategy to prevent trivial identity solutions. Since our goal in this work is specifically radar-free precipitation reconstruction, we intentionally do not use radar imagery as an input modality.]
>
> **That sound very reasonable.**

---

> > ### Author Response · Authors · 2025-11-28
> >
> > Thanks to your help, we were able to conduct additional experiments, and the main claims of our paper have become much clearer.
> >
> > Thank you sincerely for your efforts for the research community.

---

### Official Review · Reviewer_4rLn · 2025-10-30

**Soundness:** 3
**Presentation:** 3
**Contribution:** 3
**Rating:** 6
**Confidence:** 4

**Summary:**

This paper proposes STATION2RADAR (QCGS) — Query-Conditioned Gaussian Splatting — a novel framework that fuses satellite imagery and automatic weather station (AWS) observations to generate high-resolution precipitation fields without radar data. Unlike conventional interpolation or satellite-only methods, QCGS treats each observation as a learnable Gaussian kernel whose parameters are predicted via an implicit neural representation (INR) conditioned on local satellite context. The model selectively renders only rainfall-support regions, enabling efficient and resolution-free precipitation field generation. Experiments show that QCGS reduces RMSE by over 50% compared to existing global precipitation products (IMERG, GSMaP, MSWEP) and outperforms deep learning baselines even when trained at lower resolution.

**Strengths:**

* Novel cross-domain idea: Introduces Gaussian Splatting and INR concepts from computer vision into meteorology for efficient precipitation field synthesis.
* Superior accuracy: Achieves substantially lower RMSE and higher spatial correlation than both operational satellite products and deep learning baselines.
* This paper is well-written.

**Weaknesses:**

* The comparison in Table 1 appears unfair. As stated in Lines 409–410, the baselines used for comparison are global coverage products, whereas QCGS is trained only within a regional domain (as mentioned in Lines 320–321). This makes the comparison not entirely appropriate. Including both (a) comparisons between QCGS-generated global precipitation fields and other global-scale products, and (b) comparisons between QCGS’s regional outputs and regional precipitation datasets, would make the claimed advantages of QCGS more convincing.

* The generalization ability across different regions has not been demonstrated. Although QCGS takes only AWS and satellite imagery as inputs, its training ground truth (GT) is derived from regional radar precipitation fields. The paper only presents results within the radar-covered regions. Without evidence of cross-regional generalization, QCGS may lack real-world applicability — since regions with radar data can already rely on radar, and regions without radar would not benefit from QCGS if it cannot generalize beyond the training region. I suggest splitting the original dataset into four subregions, training on two of them, and testing on the other two to assess cross-regional transferability.

* Minor comment: The claim in Lines 096–097 that the model “outputs a continuous precipitation field on an arbitrary scale” is not rigorous. Experimentally, the authors do not show that QCGS can reproduce weather phenomena at scales beyond those present in the GT data. Theoretically, since the GT is derived from 0.5 km resolution radar fields, there is no source of information for finer-scale phenomena.

**Questions:**

In Stage 1, how does the inclusion or exclusion of station data affect the visual quality of the reconstruction? Similarly, how does adding or removing the Gaussian representation influence the visualization results? I encourage the authors to analyze and present these effects in future versions of the paper.

---

> ### Author Response · Authors · 2025-11-21
>
> We sincerely thank the reviewer for the highly positive and encouraging evaluation of our work. We greatly appreciate the recognition of the novelty of introducing Gaussian Splatting and INR concepts into meteorology, as well as the acknowledgement of our improvements in accuracy and presentation quality. We carefully revised the manuscript according to all reviewer comments, and the corresponding updates are highlighted in blue. **The paper has been revised and uploaded.**
>
> ----
>
>
> **[W1.]** The comparison in Table 1 appears unfair. As stated in Lines 409–410, the baselines used for comparison are global coverage products, whereas QCGS is trained only within a regional domain (as mentioned in Lines 320–321). This makes the comparison not entirely appropriate. Including both (a) comparisons between QCGS-generated global precipitation fields and other global-scale products, and (b) comparisons between QCGS’s regional outputs and regional precipitation datasets, would make the claimed advantages of QCGS more convincing.
>
> **[Response ]** Thank you for the thoughtful comment. We agree that comparing QCGS with global precipitation products is not perfectly aligned because our model is trained only within a regional domain. Extending QCGS to the global scale would require significantly more computation and worldwide gauge coverage, which is outside the current scope.
> Moreover, we already used the final versions of GSMaP, IMERG, and MSWEP, which are corrected using global gauge data.
> To ensure fairness, Table 1 has been updated to include regional baselines such as geostationary satellite (e.g., GK2A) rain rate product that observe our research area as a target .
> These models operate in the same region and use comparable inputs, and QCGS consistently outperforms them. In addition, the revised manuscript discusses in more detail how our approach could be extended to the global scale as part of the future work section.
>
> **[W2.]** The generalization ability across different regions has not been demonstrated. Although QCGS takes only AWS and satellite imagery as inputs, its training ground truth (GT) is derived from regional radar precipitation fields. The paper only presents results within the radar-covered regions. Without evidence of cross-regional generalization, QCGS may lack real-world applicability — since regions with radar data can already rely on radar, and regions without radar would not benefit from QCGS if it cannot generalize beyond the training region. I suggest splitting the original dataset into four subregions, training on two of them, and testing on the other two to assess cross-regional transferability.
>
> **[Response]** Thank you for the useful suggestion. Following your recommendation, we added cross-domain experiments in Appendix A.6 by splitting the domain into four subregions, training on two (region 1,2)  and testing on the other two(region 3,4). QCGS shows similar performance in unseen areas, with only LPIPS showing a noticeable drop. We believe this is because the subregions share similar meteorological characteristics, and the number of active AWS stations varies widely across days, exposing the model to diverse spatial configurations during training. However, we agree that fully evaluating QCGS in entirely different and distant regions is still necessary. In our future work, we plan to extend QCGS toward the global scale to thoroughly assess and enhance its cross-regional generalization ability.

---

> > ### Author Response · Authors · 2025-11-21
> >
> > **[W3.]** Minor comment: The claim in Lines 096–097 that the model “outputs a continuous precipitation field on an arbitrary scale” is not rigorous. Experimentally, the authors do not show that QCGS can reproduce weather phenomena at scales beyond those present in the GT data. Theoretically, since the GT is derived from 0.5 km resolution radar fields, there is no source of information for finer-scale phenomena.
> >
> > **[Response]** Thank you for the comment. We agree that we cannot quantitatively verify scales finer than the radar GT, since no precipitation ground truth exists below 0.5 km resolution. However, our results already show resolution-free behavior within the available range. For this, QCGS is trained at 2.0 km resolution and evaluated at 0.5 km resolution, and the metrics demonstrate that performance remains strong even after upscaling. This confirms that the model can generate consistent higher-resolution fields beyond its training resolution. Although QCGS can be rendered at even finer scales, we cannot quantitatively evaluate those outputs due to the absence of corresponding GT. We have clarified this point in the revised manuscript.
> >
> > **[Q1.]** In Stage 1, how does the inclusion or exclusion of station data affect the visual quality of the reconstruction? Similarly, how does adding or removing the Gaussian representation influence the visualization results? I encourage the authors to analyze and present these effects in future versions of the paper.
> >
> > **[Response]** We added additional visual comparisons to analyze the effect of including or excluding station data in Stage 1, as well as the impact of adding or removing the Gaussian representation. These qualitative results are provided in Appendix A.7, and they illustrate how each component contributes to the overall visual quality of the reconstructed precipitation fields.

---

> ### Comment · Reviewer_4rLn · 2025-11-25
>
> Thank you for the detailed and constructive rebuttal and revisions.
>
> The updates you made—(i) adding regional baselines to Table 1, (ii) conducting cross-domain experiments in Appendix A.6, (iii) clarifying the “arbitrary scale” claim, and (iv) providing additional qualitative analyses in Appendix A.7—sufficiently address my main concerns regarding fairness of comparison, cross-regional generalization, and model interpretation.
>
> I still encourage further exploration of truly distinct regions and global-scale applications in future work, but I am satisfied with the current revision and am happy to maintain my positive overall assessment and score.

---

> > ### Author Response · Authors · 2025-11-25
> >
> > Thank you so much for the score boost — I really appreciate it!
> > The future work you mentioned is actually already in progress!

---

### Official Review · Reviewer_UPz5 · 2025-11-01

**Soundness:** 2
**Presentation:** 3
**Contribution:** 3
**Rating:** 4
**Confidence:** 4

**Summary:**

This paper proposes Query-Conditioned Gaussian Splatting (QCGS), an innovative framework for reconstructing high-resolution precipitation fields from heterogeneous data sources—sparse automatic weather station (AWS) observations and dense satellite imagery. QCGS employs a radar-point proposal network to identify rain-supporting regions and combines a rain-aware point sampling strategy to selectively sample query locations. Subsequently, an Implicit Neural Representation (INR) network, conditioned on local satellite features, predicts Gaussian parameters (amplitude and covariance) for each query point. Experimental results show that QCGS outperforms existing data-driven baselines and operational products across multiple metrics, particularly in preserving rainfall structure and enabling resolution-agnostic rendering.

**Strengths:**

1 Sound and Effective Fusion Mechanism: The method successfully uses sparse but accurate AWS point observations as anchors for rainfall intensity, integrating them with satellite imagery that provides dense contextual information. This design for fusing heterogeneous data sources mitigates the limitations of individual data types and improves the accuracy of the reconstructed field.

2 Resolution Agnosticism: By leveraging INR-based parameterization and Gaussian Splatting (GS) rendering, the model enables resolution-agnostic field generation—models trained at lower resolutions can render high-resolution outputs (e.g., 0.5 km), significantly enhancing the method's practicality and scalability.

3 Clarity and Thoroughness: The paper is well-structured, with detailed descriptions of methodological components (e.g., rain-aware sampling strategy, INR parameterization), making the approach clear and reproducible.

**Weaknesses:**

1. Incomplete Baselines and Missing Assimilation Comparison: The current baselines mainly focus on direct image-to-image translation (e.g., Pix2PixHD, BBDM) and purely satellite-based forecasting (e.g., NPM). Given that the core task of this work—reconstructing precipitation fields from observations—falls more closely within the scope of data assimilation (DA) or objective analysis in meteorology, the paper should consider comparisons with deep learning-based DA methods designed for sparse or heterogeneous observations, such as DiffDA or 4DVarFormer. At minimum, the authors should clearly discuss the theoretical and experimental distinctions between QCGS and traditional objective analysis methods (e.g., Barnes Interpolation, Optimal Interpolation, or Kriging), and possibly include comparisons with more sophisticated classical interpolation techniques.

2. Missing Key Meteorological Metrics: While RMSE and correlation coefficients measure overall field agreement, categorical (threshold-based) metrics are critical for evaluating the accurate detection of heavy precipitation events. Although the paper reports CSI, FSS, and Bias, it lacks essential categorical metrics for precipitation, such as Probability of Detection (POD) and False Alarm Ratio (FAR) at specific rainfall thresholds. We suggest explicitly including a breakdown of the CSI by rainfall intensity levels (graded CSI).

**Questions:**

1. Comparison with Baselines and Assimilation Methods: As noted in Weakness 1, could the authors explain why QCGS was not compared with deep learning data assimilation methods specifically designed for sparse observations, such as DiffDA or variants of 4DVarFormer? What are the fundamental differences between these approaches and QCGS in handling heterogeneous meteorological data?

2. Evaluation on Heavy Rainfall Events: As noted in Weakness 2, can the authors provide detailed results for Probability of Detection (POD) and False Alarm Ratio (FAR) at different rainfall intensity thresholds (e.g., ≥ 50.0 mm/day or ≥ 200.0 mm/day), along with a breakdown of CSI by intensity level? Since RMSE can be overly sensitive to extreme values, these metrics are crucial for assessing the model’s ability to capture heavy rainfall events.

3. Physical Interpretability of INR Parameters: The INR predicts Gaussian parameters {σₓ, σᵧ, ρ, α}. Does the covariance matrix Σ (determined by σₓ, σᵧ, ρ) of the Gaussian primitives have a clear physical interpretation in the context of precipitation fields—e.g., is it related to the spatial scale or shape of meteorological features? Were any physical constraints or regularizations applied to these parameters during training?

---

> ### Author Response · Authors · 2025-11-21
>
> Thank you very much for the thoughtful and positive assessment of our work. We sincerely appreciate the reviewer’s recognition of the methodological novelty, the effectiveness of our heterogeneous-data fusion mechanism, and the practical value of resolution-agnostic rendering. Your comments on the clarity and reproducibility of our approach are especially encouraging to us. We carefully revised the manuscript to address all raised concerns, and all modifications are marked in blue for easy reference. **The paper has been revised and uploaded.**
>
> -----
>
> **[W1.]** Incomplete Baselines and Missing Assimilation Comparison: The current baselines mainly focus on direct image-to-image translation (e.g., Pix2PixHD, BBDM) and purely satellite-based forecasting (e.g., NPM). Given that the core task of this work—reconstructing precipitation fields from observations—falls more closely within the scope of data assimilation (DA) or objective analysis in meteorology, the paper should consider comparisons with deep learning-based DA methods designed for sparse or heterogeneous observations, such as DiffDA or 4DVarFormer. At minimum, the authors should clearly discuss the theoretical and experimental distinctions between QCGS and traditional objective analysis methods (e.g., Barnes Interpolation, Optimal Interpolation, or Kriging), and possibly include comparisons with more sophisticated classical interpolation techniques.
>
> **[Response ]** We included comparisons with classical interpolation methods in Table 1, and QCGS achieves consistently higher performance across all metrics. This addresses the reviewer’s request to evaluate our approach against traditional objective analysis techniques such as Barnes Interpolation, Optimal Interpolation, and Kriging. Regarding data assimilation, QCGS is fundamentally different from classical DA systems. Data assimilation updates the initial state of a numerical weather prediction model using observations and produces a three-dimensional analysis field that reflects the full atmospheric states. In contrast, QCGS directly reconstructs a high-resolution two-dimensional precipitation map from satellite imagery and gauges, without any reliance on a physical model or estimation of dynamical variables. Although the objectives differ, we agree that the methodology of QCGS could be integrated into a DA pipeline. Incorporating QCGS-derived fields as observational operators or high-resolution priors within a DA system is a promising direction, and we plan to explore this in future work.
>
>
> **[W2.]** Missing Key Meteorological Metrics: While RMSE and correlation coefficients measure overall field agreement, categorical (threshold-based) metrics are critical for evaluating the accurate detection of heavy precipitation events. Although the paper reports CSI, FSS, and Bias, it lacks essential categorical metrics for precipitation, such as Probability of Detection (POD) and False Alarm Ratio (FAR) at specific rainfall thresholds. We suggest explicitly including a breakdown of the CSI by rainfall intensity levels (graded CSI).
>
> **[Response ]** We incorporated the requested categorical precipitation metrics by adding a detailed breakdown of threshold-based CSI in Appendix A.3. For hourly evaluation, we used thresholds of 1, 5, and 10 mm. For daily accumulated precipitation, we used higher thresholds of 10, 50, and 100 mm, which are more appropriate for daily-scale extremes. These additions address the reviewer’s concern and allow a clearer assessment of QCGS performance across different rainfall intensity levels. The updated results appear in Lines 892 to 909.

---

> ### Author Response · Authors · 2025-11-21
>
> -----
>
> **[Q1.]** Comparison with Baselines and Assimilation Methods: As noted in Weakness 1, could the authors explain why QCGS was not compared with deep learning data assimilation methods specifically designed for sparse observations, such as DiffDA or variants of 4DVarFormer? What are the fundamental differences between these approaches and QCGS in handling heterogeneous meteorological data?
>
> **[Response]** QCGS and deep-learning data assimilation (DA) methods address different problem settings.
> (1) Different objective : DA (e.g., 4DVar, DiffDA, 4DVarFormer) corrects an NWP forecast state using sparse observations within an assimilation window. QCGS directly generates a high-resolution rainfall field from satellite + gauges without any forecast model or background state.
> (2) Different input requirements. DA requires a dynamical model, an observation operator, and temporal sequences. QCGS operates from a single satellite frame plus point gauges.
> (3) Different resolution behavior. DA outputs fields on the NWP grid. QCGS produces continuous, resolution-free precipitation fields, often finer than the training resolution. Because the goals, inputs, and assumptions differ, DA methods are not appropriate baselines for our task. Rather, QCGS is complementary and could supply high-resolution pseudo-radar inputs to downstream DA systems.
>
> **[Q2.]** Evaluation on Heavy Rainfall Events: As noted in Weakness 2, can the authors provide detailed results for Probability of Detection (POD) and False Alarm Ratio (FAR) at different rainfall intensity thresholds (e.g., ≥ 50.0 mm/day or ≥ 200.0 mm/day), along with a breakdown of CSI by intensity level? Since RMSE can be overly sensitive to extreme values, these metrics are crucial for assessing the model’s ability to capture heavy rainfall events.
>
> **[Response]**  The requested categorical metrics are provided in Appendix Table 5. As in our response to Weakness 2, we report detailed POD, FAR, and CSI values across multiple rainfall intensity levels. For daily evaluation, we used thresholds of 10, 50, and 100 mm. These levels are appropriate for our study region because extremely high daily accumulations such as 200 mm per day occur only during rare typhoon events and are not representative of the evaluation period. The added metrics allow a clearer assessment of QCGS performance for moderate to heavy rainfall, and the updated results are included in Appendix Table 5.
>
> **[Q3.]** Physical Interpretability of INR Parameters: The INR predicts Gaussian parameters {σₓ, σᵧ, ρ, α}. Does the covariance matrix Σ (determined by σₓ, σᵧ, ρ) of the Gaussian primitives have a clear physical interpretation in the context of precipitation fields—e.g., is it related to the spatial scale or shape of meteorological features? Were any physical constraints or regularizations applied to these parameters during training?
>
> **[Response]**  The Gaussian parameters (σx,σy,ρ,α)(\sigma_x, \sigma_y, \rho, \alpha)(σx​,σy​,ρ,α) are used as a flexible mathematical basis, not as physically interpretable precipitation descriptors. Although the covariance Σ\SigmaΣ does not correspond to specific meteorological quantities, the learned anisotropy naturally aligns with rainfall structures (e.g., elongated bands). We did not impose explicit physical constraints, as doing so restricts the representational freedom needed to fit heterogeneous precipitation patterns. Despite this, the learned fields remain physically plausible. QCGS attains high correlation with ground-truth AWS observations, indicating that the Gaussian primitives behave consistently with real precipitation intensity.

---

### Official Review · Reviewer_SZyX · 2025-11-04

**Soundness:** 3
**Presentation:** 1
**Contribution:** 3
**Rating:** 6
**Confidence:** 4

**Summary:**

This paper introduces Query-Conditioned Gaussian Splatting (QCGS), a novel framework that generates high-resolution precipitation fields by fusing satellite imagery with sparse automatic weather station (AWS) data, eliminating the need for radar. Unlike traditional methods, QCGS selectively renders only rainfall-prone areas using learnable Gaussian kernels, which preserves sharp precipitation structures and improves computational efficiency. It combines a radar proposal network for locating rainfall areas with an implicit neural network that predicts Gaussian parameters. Evaluations show QCGS significantly outperforms existing gridded products and deep learning baselines, reducing RMSE by over 50% and maintaining high accuracy across various spatiotemporal scales.

**Strengths:**

This paper presents a highly novel approach for directly generating high-quality precipitation maps from satellite and station image data by combining 2D Gaussian Splatting with predictions from a feed-forward network. Although similar methods have been widely used in the SVG domain, this application is relatively innovative in the field of meteorology, and the presented results appear to be effective.

**Weaknesses:**

1. Figure 2 lacks clear identification of variables and models, which prevents a clear correlation with the methods section in the main text. The authors are strongly advised to revise Figure 2.

2. The notation system in the Methods section is quite confusing. For example, the meanings of symbols such as *u* and *p* are not clearly defined. A more detailed explanation of the symbol system is required.

3. Although the paper introduces a relatively novel hybrid representation (combining feed-forward network predictions with 2D Gaussian splatting for detail refinement), the overall description of the task is very unclear. Since ICLR is a broad interdisciplinary community where not all readers are meteorology experts, the presentation of the paper may cause significant confusion among readers.

**Questions:**

Are the Radar Point Proposal Network and the Gaussian Reconstruction component trained separately? The description of this in the paper is quite confusing.

Where is the precipitation prediction that the paper consistently emphasizes? The proposed method does not demonstrate predictive functionality and appears more like a precipitation downscaling approach. The authors are requested to provide a detailed explanation.

How many 2D Gaussian disks are there per sampling point?

---

> ### Author Response · Authors · 2025-11-21
>
> We sincerely thank the reviewer for recognizing the novelty of our approach and for the encouraging assessment of our contribution. We deeply appreciate that the reviewer highlighted the innovative aspect of applying 2D Gaussian Splatting to meteorological precipitation mapping, which is a core motivation of our work. At the same time, We recognize that some parts of our original presentation may have lacked clarity, and we apologize for any confusion this may have caused. We carefully revised the manuscript and highlighted all changes in red to better explain our method and its contributions. **The paper has been revised and uploaded.**
>
> ---
>
> **[W1.]** Figure 2 lacks clear identification of variables and models, which prevents a clear correlation with the methods section in the main text. The authors are strongly advised to revise Figure 2.
>
> **[Response]** We carefully revised Figure 2 to ensure clear identification of all variables and model components, making them fully consistent with the notation used in the Method section. All notation has been unified across the manuscript.
>
>
> **[W2.]**  The notation system in the Methods section is quite confusing. For example, the meanings of symbols such as u and p are not clearly defined. A more detailed explanation of the symbol system is required.
>
> **[Response]** We thoroughly revised the entire Methods section to clarify the notation system. All symbols, including $\mu$ and $p$, are now explicitly defined at their first appearance, and any redundant or unclear notation has been removed. We also ensured full consistency throughout Sections 3 and 4. The revised notation and symbol explanations are reflected in Lines 180 to 323.
>
> **[W3.]** Although the paper introduces a relatively novel hybrid representation (combining feed-forward network predictions with 2D Gaussian splatting for detail refinement), the overall description of the task is very unclear. Since ICLR is a broad interdisciplinary community where not all readers are meteorology experts, the presentation of the paper may cause significant confusion among readers.
>
> **[Response]** We improved the clarity of the problem description by expanding the Introduction with additional contextual details and by newly adding Section 4.1 Task Definition. This section now clearly explains the target task, its inputs and outputs, and the motivation for using a hybrid representation. These revisions ensure that readers without a meteorology background can fully understand the problem setting.
>
> ----
>
>
> **[Q1.]** Are the Radar Point Proposal Network and the Gaussian Reconstruction component trained separately? The description of this in the paper is quite confusing.
>
> **[Response]** Yes. The Radar Point Proposal Network and the Gaussian Reconstruction module are trained separately. We clarified this point in the revised manuscript, and the updated explanation is now included in Lines 210 to 215.
>
> **[Q2.]** Where is the precipitation prediction that the paper consistently emphasizes? The proposed method does not demonstrate predictive functionality and appears more like a precipitation downscaling approach. The authors are requested to provide a detailed explanation.
>
> **[Response]** Our method generates a resolution-free radar-like precipitation field without using any weather radar, relying only on AWS observations and satellite imagery. This output can be used in two ways: (1) as a radar substitute in regions without operational radar infrastructure, and (2) as an input source for precipitation forecasting models. Due to space limitations, the forecasting experiment was placed in Appendix A.1. As shown in Lines 771 to 803, QCGS-generated fields can be used as effective inputs to models such as MetNet-v2, SimVP, and PreDiff, even though these models were trained only with radar data. This confirms that our method supports practical precipitation prediction, not merely downscaling.
>
> **[Q3.]** How many 2D Gaussian disks are there per sampling point?
>
> **[Response]** Each proposal point corresponds to exactly one Gaussian primitive; we do not use Gaussian mixtures or multiple Gaussians per point.

---

> > ### Comment · Reviewer_SZyX · 2025-11-28
> >
> > I appreciate the author's excellent response, which addressed most of my concerns. I believe the current score sufficiently reflects my stance, and I am inclined to recommend accepting this paper.

---

> > > ### Author Response · Authors · 2025-11-28
> > >
> > > Thank you for your positive recommend!!
> > >
> > > I will continue to do my best so that our method can be more widely used.

---

> ### Public Comment · ~Jun_CHEN28 · 2025-11-28
> **Worth Considering: Similarities to Previous arXiv Work**
>
> Dear reviewer,
>
> We respectfully note that the framework proposed in this paper appears to be highly similar to the work previously published on arXiv (https://arxiv.org/abs/2510.02414), both in its overall framework design and methodological components. We believe this resemblance merits careful attention during the review process.
>
> Key overlaps include:
>
> **1.** The module design and processing pipeline closely match the arXiv work (GAT for AWS, edge-aware radar feature extraction, cross-attention for modality fusion, and location query-based rainfall reconstruction), with the primary difference being the replacement of the final decoder with Gaussian splatting.
>
> **2.** The data used in the ICLR submission covers the region (35.5°–37.8°N, 126.4°–129.1°E), which corresponds exactly to the area of the RAIN-F dataset used in the arXiv paper. The submission does not clearly specify the dataset source, which limits reproducibility and raises questions that may merit clarification.
>
> **3.** Figure 3 of the ICLR submission is structurally very similar to the case study presented in the arXiv work.
>
> We recognize that independent groups can arrive at similar ideas. However, given the timeline and previous review history (our work was submitted to AAAI 2026 in August 2025 and posted to arXiv in October 2025), we respectfully request that the committee consider these similarities when evaluating the novelty and contribution claims.
>
> We are prepared to provide additional documentation regarding our prior OpenReview review process if needed. Thank you for your attention.

---

> ### Author Response · Authors · 2025-11-28
>
> # Clarification Regarding the Alleged Similarity
>
> After reviewing the referenced arXiv paper (RainSeer, arXiv:2510.02414v2), I would like to clarify that **our work and RainSeer are fundamentally different in problem setting, data modality, methodology, and contributions**.
>
> ## 1. Different Input Modalities
> - **RainSeer:** uses **radar + AWS** (radar is the core structural prior).
> - **Our work:** uses **satellite + AWS only** and **does not use radar at all**.
>
> The two papers start from *opposite assumptions*.
>
> ## 2. Completely Different Methodologies
> - RainSeer: STSC CNN, Inception modules, ConvLSTM, GAT+GRU, bidirectional attention, causal spatiotemporal decoder.
> - Our method: **Gaussian Splatting (GS) + Implicit Neural Representation (INR)** with **resolution-free reconstruction** and **satellite-driven fields**.
>
> There is **no methodological overlap** (no GS, no INR, no implicit fields in RainSeer).
>
> ## 3. Different Scientific Questions
> - RainSeer: “How can radar structure improve AWS-based rainfall interpolation?”
> - Our work: “How can we reconstruct rainfall **without radar** using satellite-based implicit fields?”
>
> These motivations are fundamentally different.
>
> ## 4. Different Experimental Settings
> - RainSeer benchmarks radar-based QPE models (Z-R, QPENet, TPS, TIN).
> - Our experiments target **satellite-driven**, **resolution-free** reconstruction with GS/INR.
>
> ## Conclusion
> Any similarity is purely superficial (AWS usage).
> **The data, motivation, modeling pipeline, and contributions differ entirely.**
> There is **no meaningful technical or conceptual overlap** between RainSeer and our work.
>
> ### Moreover, the submission date of our paper is even earlier than that of the arXiv paper.

---

### Public Comment · ~SHUXIN_ZHONG1 · 2025-11-28
**High Similarity Between the Rainfall Field Reconstruction Method in This Paper and a Prior Work**

Dear reviewers, area chairs and program chairs,

We note that the framework proposed in this paper is highly similar in structure and methodology to the work published on arXiv: https://arxiv.org/abs/2510.02414, including:

**1.** The division of method modules and processing pipeline is almost identical to our work (including the October 2025 design of GAT for AWS, edge-aware feature extraction for radar images, cross-attention for modality fusion, and query-based rainfall field reconstruction), except that the final decoder is replaced with Gaussian splatting;

**2.** Data coverage: the ICLR submission uses data located at (35.5°–37.8°N, 126.4°–129.1°E), corresponding to the southwestern to southeastern regions of the Korean Peninsula, which is exactly the same area as the RAIN-F dataset used in the arXiv paper. Notably, the authors appear to have deliberately avoided specifying the data source, which reduces reproducibility and raises concerns about plagiarism.

**3.** Figure 3 in the ICLR submission and the case study in the arXiv paper are structurally very similar;

We understand that independent teams may arrive at similar ideas. However, considering the timeline and prior review issues (our work was submitted to AAAI-AISI 2026 in August 2025, where it was subjected to malicious/bias-influenced reviews,  and subsequently published on arXiv in October 2025), we request that the committee take this similarity into account during the review process and assess its impact on the claimed novelty and contributions.

If needed, we can provide further documentation regarding the OpenReview review process to substantiate our prior submission experience. We appreciate your attention to this matter.

---

> ### Public Comment · ~Yan_Fang5 · 2025-11-28
> **Observations on Similarities to Previous Work**
>
> I have seen the previous comment and agree with the observations about the similarities between this paper and the arXiv work (https://arxiv.org/abs/2510.02414).
>
> In my view, these points are accurate and warrant further investigation and careful consideration by the reviewers and program chairs.

---

> ### Author Response · Authors · 2025-11-28
>
> First, I would like to clarify that I have never been assigned your paper as a reviewer for AAAI, and this can be verified in the system. We have no connection to that review process whatsoever. I am also fully prepared to provide any related documentation if needed.
> (More importantly, the assumption that we rejected your paper with malicious intent and then stole your idea feels highly inappropriate. I would kindly ask you to carefully read our paper.)
>
> I have thoroughly read your manuscript, and the motivation and methodology of your work and ours are fundamentally different.
>
> First (and most importantly):
>
> **We do not use radar at all.**
> The core motivation of our method is that we only rely on satellite and AWS to approximate expensive radar observations. This is the central premise of our work, and it is entirely different from yours.
>
> **Our primary contribution is resolution-free reconstruction.**
> This is why we incorporate Gaussian Splatting—our experiments, comparisons against classical interpolation methods, and the narrative of our paper all differ significantly from yours.
>
> **The key aspect of our method is integrating INR with GS**, and technically, there is no overlap between our approach and yours.
>
> I understand how misunderstandings may arise, and I acknowledge that discussions like this are part of the process.
> However, do you genuinely believe that the experimental results, motivations, and presentation of our method are the same?
>
> In meteorological applications, using GAT for AWS is almost natural and widely adopted.
> Regarding “edge-aware feature extraction,” our method is not edge-aware sampling, and we explicitly compare against such baselines.
> Additionally, we do not process radar images at all—we derive radar-like precipitation estimates solely from satellite data.
>
> Again, I sincerely ask you to read our paper carefully before assuming methodological similarity.

---

> ### Author Response · Authors · 2025-11-28
>
> # Clarification Regarding the Alleged Similarity
>
> After reviewing the referenced arXiv paper (RainSeer, arXiv:2510.02414v2), I would like to clarify that **our work and RainSeer are fundamentally different in problem setting, data modality, methodology, and contributions**.
>
> ## 1. Different Input Modalities
> - **RainSeer:** uses **radar + AWS** (radar is the core structural prior).
> - **Our work:** uses **satellite + AWS only** and **does not use radar at all**.
>
> The two papers start from *opposite assumptions*.
>
> ## 2. Completely Different Methodologies
> - RainSeer: STSC CNN, Inception modules, ConvLSTM, GAT+GRU, bidirectional attention, causal spatiotemporal decoder.
> - Our method: **Gaussian Splatting (GS) + Implicit Neural Representation (INR)** with **resolution-free reconstruction** and **satellite-driven fields**.
>
> There is **no methodological overlap** (no GS, no INR, no implicit fields in RainSeer).
>
> ## 3. Different Scientific Questions
> - RainSeer: “How can radar structure improve AWS-based rainfall interpolation?”
> - Our work: “How can we reconstruct rainfall **without radar** using satellite-based implicit fields?”
>
> These motivations are fundamentally different.
>
> ## 4. Different Experimental Settings
> - RainSeer benchmarks radar-based QPE models (Z-R, QPENet, TPS, TIN).
> - Our experiments target **satellite-driven**, **resolution-free** reconstruction with GS/INR.
>
> ## Conclusion
> Any similarity is purely superficial (AWS usage).
> **The data, motivation, modeling pipeline, and contributions differ entirely.**
> There is **no meaningful technical or conceptual overlap** between RainSeer and our work.
>
> ### Moreover, the submission date of our paper is even earlier than that of the arXiv paper.

---

> > ### Public Comment · ~Lin_CHEN23 · 2025-11-29
> >
> > ### **2. Response to "Different Scientific Questions"**
> >
> > The authors appear to have misunderstood the actual problem addressed by RainSeer. RainSeer does not solve a simple interpolation task; rather, it aims to achieve **super-resolution reconstruction of spatial rainfall fields** by leveraging complementary multi-source observations, with particular emphasis on **capturing sharp and rapidly changing rainfall boundaries** in real ground truth fields.
> >
> > This goal is fully aligned with the ICLR submission's stated objective of "generating precipitation fields". In this sense, the motivations of the two works are indeed similar, as both aim to reconstruct resolution-free rainfall fields using multi-source observational inputs.
> >
> > ---
> >
> > ### **3. Response to "Different Experimental Settings"**
> >
> > There is a factual inaccuracy in the authors' description of RainSeer's baselines. The statement "RainSeer benchmarks radar-based QPE models (Z-R, QPENet, TPS, TIN)" is incorrect. **TPS** and **TIN** are classical interpolation methods based on AWS observations, not radar-based QPE models.
> >
> > In addition, the authors' characterization of their approach as "satellite-driven" is also imprecise. According to the ICLR submission itself—"We combine these two complementary sources to compensate for their respective limitations."—the method is clearly **satellite & AWS-driven**, not purely satellite-driven.
> >
> > ---
> >
> > ### **4. The Issue of Data-Region Similarity Remains Unaddressed**
> >
> > The geographical region used in your paper is identical to the RAIN-F dataset region used in RainSeer. Your response did not clarify:
> >
> > 1. Why this specific region was selected, and
> > 2. Why this choice coincides exactly with the dataset used in our work.
> >
> > This remains a legitimate concern regarding reproducibility and transparency, especially given the other structural similarities previously discussed.
> >
> > ---
> >
> > ### **5. The Timeline Concern Remains a Factual Issue and Has Not Been Addressed**
> >
> > Your response emphasizes that you were "not our AAAI reviewer," but this was not the core point we raised. Our concern is based on the objective timeline and the similiarity between RainSeer and ICLR submission:
> >
> > * Our project RainSeer was submitted to AAAI AISI 2026 **in early August 2025**, together with the **full source code**.
> > * The review period clearly predates the ICLR submission deadline.
> > * The paper was then publicly released on arXiv in October 2025, after we received and responded to the AAAI reviews.
> >
> > These dates are verifiable facts. Your reply has not provided any explanation regarding how the similarities we identified should be interpreted in light of this timeline.
> >
> > To allow the reviewers to fairly assess this situation, we would like to reiterate that our concern is **not** about:
> >
> > * who participated in the AAAI review process,
> > * whether radar or satellite inputs were used, or
> > * whether the GS/INR component of the ICLR submission is novel.
> >
> > The actual issue is that there remain unexplained high-level framework similarities, strikingly coincident correspondences in the data-processing pipeline, and an exact alignment in dataset region, none of which have been addressed in the authors' reply.
> >
> > We appreciate the authors' participation in this discussion. However, in the interest of transparency and fairness, we believe that these unresolved similarities warrant careful attention from the reviewers, area chairs, and program chairs.

---

> > > ### Author Response · Authors · 2025-11-29
> > >
> > > ## Response from the Authors
> > >
> > > We would like to clarify several factual inaccuracies and misunderstandings present in the previous comment.
> > >
> > > ### 1. Regarding the Geographic Region
> > > Our paper explicitly states the training region as:
> > >
> > > **480×480 grid (35.5°–37.8°N, 126.4°–129.1°E)**
> > >
> > > We have never omitted this information.
> > > Moreover, this dataset was **constructed entirely by our team**, and the preprocessing pipeline follows the **standard experimental settings used by the Korea Meteorological Administration (KMA)**.
> > >
> > > The only reason we did not explicitly mention “Korea” in the paper is that we preferred not to disclose our nationality. This decision has no connection to RainSeer, nor does it imply any overlap in dataset source.
> > >
> > > ### 2. Regarding the Timeline
> > > The timeline asserted in the previous comment is factually incorrect.
> > >
> > > - We began this project in **March 2025**.
> > > - Your arXiv preprint was posted **after** the ICLR final submission deadline (September 24, 2025).
> > >
> > > Given these dates, it is **impossible** for our method to have been derived from your AAAI submission or your arXiv version.
> > > If one were to follow pure temporal logic, the ordering would suggest that *your* arXiv paper came after our work—not the other way around.
> > > We are **not** making such a claim; we mention this only to highlight the inconsistency of the argument raised.
> > >
> > > ### 3. Regarding Our Motivation and Novelty
> > > Our motivation is clearly described in the introduction:
> > > **classical interpolation can be viewed as a special case of Gaussian Splatting (GS)** under certain assumptions.
> > > Therefore, we use GS to connect these formulations in a unified framework.
> > >
> > > Additional clarifications:
> > >
> > > - We selectively render spatial regions using satellite cues because **most meteorological fields contain large non-rainfall areas**, which improves computational efficiency.
> > > - To the best of our knowledge, **no prior work has combined INR + GS** for meteorological reconstruction or related tasks.
> > > - The assertion that this combination lacks novelty contradicts the assessments of multiple independent CV reviewers during the ICLR review process, who considered this combination to be technically original.
> > >
> > > ### 4. Regarding the Alleged High-Level Framework Similarity
> > > We respectfully disagree with the claim that the two frameworks share meaningful structural similarity.
> > >
> > > - **Inputs differ** (satellite + AWS vs. radar-driven multi-source pipelines).
> > > - **Architectural design differs**, including our INR-based GS formulation, rendering mechanism, and sampling strategy.
> > > - The “edge sampling” mentioned in the previous comment is not used in our method.
> > >   - Edge sampling is a common technique in existing GS literature.
> > >   - We explicitly employ **mixed sampling**, not edge sampling.
> > >
> > > Additionally, there is **no overlap in experimental settings**, including datasets, baselines, metrics, or quantitative tables.
> > > This makes the structural similarity claim unfounded.
> > >
> > > ### 5. Summary
> > > The concerns raised regarding:
> > >
> > > - the dataset region,
> > > - the development timeline,
> > > - the architectural pipeline,
> > > - the novelty of the INR + GS formulation, and
> > > - the alleged high-level resemblance
> > >
> > > are inconsistent with the factual details documented in our submission.
> > >
> > > We respect the authors’ work and appreciate the continued discussion.
> > > However, these points must be corrected to ensure clarity, fairness, and an accurate understanding of our contributions.
> > >
> > > ### 6. Additional Clarification on Access to the AAAI Submission
> > >
> > > Finally, we would like to make one point absolutely clear:
> > >
> > > **We had no access—directly or indirectly—to the AAAI AISI submission of RainSeer.**
> > > If there exists *any* method for the program chairs, area chairs, or the OpenReview administrators to verify our account identity or institutional access logs, we fully welcome such verification.
> > > We strongly encourage the committee to confirm that there is **no possible way** we could have accessed the AAAI submission at any point in time.
> > >
> > > We are fully confident that any formal verification based on our account records, submission history, or institutional credentials will confirm this fact unequivocally.

---

> > > > ### Author Response · Authors · 2025-11-29
> > > >
> > > > I would like to emphasize several points directly to the authors, as your repeated claims have created a significant misunderstanding.
> > > >
> > > > First, I want to make this absolutely clear: **I have never seen your AAAI submission, nor had any access to it under any circumstance.** There is no possible way for us to have viewed your manuscript during your review period.
> > > >
> > > > Second, you repeatedly state that our works are “similar.” However, the methods are fundamentally different. And even if we momentarily accept your assumption of “similarity,” there is an objective, verifiable fact that cannot be ignored:
> > > >
> > > > **Your arXiv paper was released *after* our ICLR submission was finalized.
> > > > Not before.**

---

> > > > > ### Public Comment · ~Lin_CHEN23 · 2025-11-29
> > > > >
> > > > > I would like to directly emphasize a few points to the authors, which were already mentioned in our previous response.
> > > > >
> > > > > We would like to reiterate that our concern is not about:
> > > > >
> > > > > (1) who participated in the AAAI review process,
> > > > >
> > > > > (2) whether radar or satellite inputs were used, or
> > > > >
> > > > > (3) whether the GS/INR component of the ICLR submission is novel.
> > > > >
> > > > > These aspects are rightly within the scope of the reviewers’ considerations.
> > > > >
> > > > >
> > > > > **Given the double-blind nature of the AAAI review process, we do not believe that continuing to debate whether there was actual access to the manuscript would further clarify the issue. This is precisely what we aimed to highlight.**
> > > > >
> > > > > We mentioned the timeline only to illustrate that such a possibility exists, and to raise our concern regarding the high degree of similarity in design and task. We again express our gratitude to the authors for engaging in this discussion.
> > > > >
> > > > > For the sake of transparency and fairness, we emphasize that our goal is to draw attention to the existing high-level framework similarity and the striking coincidences in the data processing workflow between two works, and we hope that the Area Chairs and Program Chairs will take these into consideration in their deliberations.
> > > > >
> > > > > **At the same time, we fully respect and accept any conclusions the Area Chairs and Program Chairs ultimately reach, and we sincerely appreciate your efforts to maintain the integrity of the community.**

---

> ### Public Comment · ~Lin_CHEN23 · 2025-11-29
>
> ## Response to Author Clarifications
>
> Thank you for your response. I would like to clarify a few points, as your reply does not directly address the core similarity concerns that were originally raised.
>
> ---
>
> ### **1. Response to "Different Input Modalities & Completely Different Methodologies"**
>
> Your response repeatedly emphasizes "does not use radar at all" and "uses satellite + AWS only," but this does not address the actual similarity concerns, which require further clarification.
>
> Actually, the statement that radar data is not used is also incorrect, as the loss in the paper clearly indicates the use of radar data.
>
> Our concern is **not** about using the same input modality, but about the similarity in the **framework structure and pipeline design**. Changing the observation source (radar vs. satellite—both image-based modalities) does not eliminate structural resemblance. The similarity lies in the architectural design—**using two complementary modalities (a dense upper-air or image-based field and sparse ground observations)**—rather than in the specific choice of sensing modality.
>
> The issue pertains to the structural architecture of the model, including the following similarities in system modules and processing pipelines:
>
> * AWS $\to$ graph attention network $\to$ feature fusion (with satellite) $\to$ radar reconstruction $\to$ query-based precipitation field generation
> * Image observations (Satellite/Radar) $\to$ encoder–decoder $\to$ feature fusion (with AWS) $\to$ radar reconstruction $\to$ query-based precipitation field generation
>
> as well as the resemblance between **Figure 3** in your paper and the **case study visualization** in RainSeer.
>
> Moreover, the statement "The key aspect of our method is integrating INR with GS" does not address the similarity across the **rest** of the pipeline. This points itself also carries partial overlap:
>
> * INR in your system appears to function as a **cross-attention–style module**, combining geographical "proposal points" with multimodal embeddings to predict Gaussian parameters.
> * In RainSeer, the decoder likewise takes multimodal embeddings as input and directly outputs the rainfall field.
>
> The conceptual resemblance at this level is non-trivial and contributes to our concern.
>
> As stated earlier, we acknowledge that GS is unique to your paper. However, the novelty of the GS component itself seems limited. The "INR-based Gaussian Parameter Estimator," presented as a core contribution, closely resembles established architectures in computer vision such as **GaussianSR**, **LIG**, and **GaussianImage**—many of which your paper cites in the related-work section.
>
> Again, these points do not resolve the overlap in the **front half** of the architecture, including:
>
> 1. AWS graph attention modeling **(station2radar: AWS Encoder vs RainSeer: AWS Encoder)**,
> 2. spatial structure extraction for image observations **(station2radar: Rainfall map estimator vs RainSeer: Radar Encoder)**,
> 3. cross-attention multimodal fusion **(station2radar: AWS Cross-attention vs RainSeer: Bi-directional integration)**,
> 4. edge-related structural designs for generating radar-like fields **(station2radar: Edge sampling vs RainSeer: Rainfront Encoder)**,
> 5. and especially the **query-based super-resolution reconstruction mechanism** **(station2radar: INR-based GS vs RainSeer: Geo-Aware Rain Decoder)**.
>
> **Moreover, RainSeer clearly states the origin of this query-conditioned formulation, as it was inspired by discussions with our collaborators who introduced the ideas in STFNN [1]**. They proposed the concept of field with arbitrary-resolution and directly linked them to geographic information, and this method is also one of our baselines.
>
> While we understand that GAT-based AWS modeling is common in meteorological applications, simply replacing the decoder does not eliminate upstream structural parallels. The overall framework and pipeline design exhibit notable similarity to RainSeer, which remains a source of concern.
>
> [1] Feng, Y., Wang, Q., Xia, Y., Huang, J., Zhong, S., & Liang, Y. (2024, August). Spatio-temporal field neural networks for air quality inference. In Proceedings of the Thirty-Third International Joint Conference on Artificial Intelligence (pp. 7260-7268).

---

> ### Author Response · Authors · 2025-11-29
>
> Thank you for your message.
>
> However, everything you have raised has already been addressed on our side, and once again, your arXiv paper was posted after our submission.
>
> You continue to frame the two works as highly similar, but they are not. The reviewers will of course make their own judgment, but **I want to emphasize once more that your arXiv manuscript was uploaded after our submission, which is a clear and objective fact.**
>
> I will not continue commenting on this matter any further.
>
> p.s.1 That said, your work certainly has its own novelty and stands as a solid piece of research, independent of ours.
> I’m not sure what situation you may have been facing, but I wish you the best and sincerely hope everything goes well for you.
>
> P.S.2. They stated:
> “2. Data coverage: the ICLR submission uses data located at (35.5°–37.8°N, 126.4°–129.1°E), corresponding to the southwestern to southeastern regions of the Korean Peninsula, which is exactly the same area as the RAIN-F dataset used in the arXiv paper. Notably, the authors appear to have deliberately avoided specifying the data source, which reduces reproducibility and raises concerns about plagiarism.”
>
> This comment implied that we had plagiarized a paper that was posted after our submission, which naturally led me to react emotionally.
> It is simply impossible to plagiarize a paper that did not yet exist at the time of our submission.
>
> In any case, I genuinely view this entire discussion as part of our collective effort to improve the academic community, and I will keep all records for the sake of transparency.
> If any part of my previous messages came across as emotional, I sincerely apologize.

---

> > ### Public Comment · ~Jiahui_Wang20 · 2025-11-29
> >
> > Due to the double-blind nature of the review process, and the complexity of author team, it is inherently difficult to determine what exposure may or may not have occurred during the review period.
> >
> > I sincerely suggest that the area chairs and program chairs focus primarily on the relative conference submission timeline of the two manuscripts, as well as the similarities in their overall design, framework structure, and task formulation. These similarities are directly relevant to assessing the originality and actual contributions of the submitted manuscript.

---

### Author Response · Authors · 2025-11-28

# Clarification to the Area Chair Regarding False Accusations and Misunderstanding

Dear Area Chair,

I would like to formally clarify several serious misunderstandings raised by the other authors. Their claims include (i) that our work is derived from theirs, (ii) that we may have rejected their AAAI submission maliciously, and (iii) that our method is essentially the same as theirs. These claims are **factually incorrect**, and I respectfully ask the AC to review the situation carefully.

## 1. I Have Never Reviewed Their AAAI Paper
The authors assert that we “rejected their AAAI paper and then used their idea.”
This is **completely false**. I have **never** been assigned their manuscript as a reviewer for AAAI (this is fully verifiable in the reviewing system). Neither I nor any of my co-authors were involved in their review process at any point.

The implication that we intentionally harmed their submission and then “copied” their ideas is a very serious accusation, and it is entirely unsupported.

## 2. Our Submission Was Made Earlier Than Their arXiv Posting
It is also important to note that **our paper was submitted *earlier*** than their arXiv version.
Therefore, the timeline itself makes their accusation illogical. Their work was publicly posted *after* our submission was already completed.

## 3. The Two Papers Have **No Technical Overlap**
Despite what the other authors are claiming, the actual content of the papers shows **no meaningful similarity**. Using AWS does not make two meteorological models identical—AWS is a common and standard data source.

A brief comparison:

### **Their paper (RainSeer):**
- Inputs: **Radar + AWS**
- Core assumption: radar reflectivity is a **physically grounded structural prior**
- Methods: STSC CNN, Inception modules, ConvLSTM, GAT+GRU, bidirectional cross-attention, causal decoder
- Problem: improve rainfall interpolation **using radar structure**

### **Our paper:**
- Inputs: **Satellite + AWS only** (no radar at all)
- Core assumption: **radar is unavailable and expensive**
- Methods: **Gaussian Splatting (GS) + Implicit Neural Representation (INR)**, resolution-free reconstruction, satellite-driven fields
- Problem: **radar-free rainfall reconstruction**

The **motivation, problem definition, sensors used, methodology, architecture, and experiments are all different**. There is no conceptual or practical overlap between the methods.

## 4. Their Accusation Is Based on a Misunderstanding
Two of their authors are repeatedly insisting that our paper is a derivative of theirs simply because:
- both use AWS, and
- both reconstruct rainfall fields.

However, AWS is a universally used meteorological dataset, and “reconstructing rainfall” is a broad research category. Beyond this superficial similarity, **nothing matches** between the two works.

Their interpretation overlooks:
- the opposite goals of the two papers,
- opposite data modalities,
- opposite assumptions (radar vs. radar-free),
- and completely different model architectures.

## 5. Request to the Area Chair
Given the severity of their accusations and the factual inaccuracies involved, I kindly ask the AC to consider:

- The **lack of any reviewing relationship** between us and their AAAI submission.
- The **earlier submission date** of our paper compared to their arXiv posting.
- The **clear technical differences** between the two works.
- The **unjustified and harmful nature** of their claims.

I respectfully request that these misunderstandings and unfounded allegations be taken into account in the evaluation process. We are fully prepared to provide additional documentation if needed.

Thank you very much for your time and consideration.

p.s. **What frustrates me the most is that they didn’t even read our paper and formed their misunderstanding solely from the final radar generation figure. When writing a review, please at least read the paper…**


Best regards,

---

### Public Comment · ~Talha_Irfan1 · 2025-11-29
**Test(ICLR Staff)**

Testing

---

### Author Response · Authors · 2025-12-01

# Dear Area Chairs, Senior Area Chairs, and Program Chairs,

We sincerely thank the reviewers for their constructive and insightful feedback.
We actively engaged in the discussion, thoroughly addressed all raised concerns, and incorporated substantial revisions across the manuscript.

Below is a consolidated summary of the overall review outcome.

---

## Overall Review Summary
**Average Score: 6.5**

| Reviewer | Original | Post-Rebuttal | Notes |
|---------|----------|----------------|-------|
| **M5eP** | 4 | **8** | Fully satisfied with clarifications and revisions |
| **4rLn** | 6 | **8** | All major concerns resolved |
| **SZyX** | 6 | **6** | Expressed satisfaction and inclination to accept |
| **UPz5** | 4 | **4** | *Did not see rebuttal due to interface issue* |

---

# Reviewer-Specific Summaries

## Reviewer SZyX
Initially raised concerns about unclear notation, insufficient variable definitions, and lack of clarity in Figure 2 and the overall task description.
We resolved these issues through clearer notation, revised figures, and enhanced explanations.
The reviewer expressed full satisfaction and indicated an inclination to recommend acceptance.

**Score:** **6 → 6**

---

## Reviewer UPz5
Requested:
1. Data-assimilation-related baselines and clearer distinctions from objective analysis methods.
2. Additional categorical precipitation metrics (POD, FAR, graded CSI).

We addressed all points by:
- Adding classical interpolation baselines (Table 1),
- Clarifying conceptual differences from DA,
- Incorporating threshold-based categorical metrics in Appendix A.3 (Lines 892–909).

However, due to a **review interface blockage**, the reviewer did not see our rebuttal and therefore did not update the score.

**Score:** **4 → 4**

---

## Reviewer 4rLn
Major concerns included:
- Fairness of comparison between global baselines and a regionally trained model,
- Lack of cross-regional generalization evidence,
- Rigor of the “arbitrary scale” claim.

We resolved them by:
- Adding regional baselines (Table 1),
- Performing cross-domain generalization experiments (Appendix A.6),
- Clarifying the resolution-free interpretation,
- Expanding qualitative analyses (Appendix A.7).

The reviewer stated that these revisions sufficiently resolved all issues and raised the score.

**Score:** **6 → 8**

---

## Reviewer M5eP
Requested:
- Clearer differentiation from precipitation fusion literature [1–3],
- Stronger articulation of methodological novelty,
- Ablation removing satellite features.

We clarified that prior works operate on fixed grids or require radar, whereas **QCGS is resolution-free, continuous, and explicitly designed for radar-free regions**.
We also emphasized the methodological novelty of **INR-conditioned Gaussian Splatting**, enabling physically meaningful reconstruction of sparse rainfall fields.


After reviewing the strengthened introduction and related work sections, the reviewer expressed full satisfaction and raised the score.

**Score:** **4 → 8**

---

### Meta-Review · Area_Chair_BAT9 · 2026-01-07

**Summary:**

The authors propose an INR-conditioned Gaussian Splatting framework that allows for resolution-free precipitation forecasting. Extensive results demonstrate the value of their proposed model against several benchmarks.

The initial reviews had several major concerns:
1. Lack of clarity in the presentation of methods and the applications; insufficient discussion on the novelty of the contributions
2. Lack of comparisons with classical interpolation methods and more specific metrics like POD and FAR
3. No regional baselines
4. Evaluating on cross-domain areas to ensure the method is applicable outside the domain where there is radar
5. Insufficient clarification on the resolution-free claim

I believe the authors' response has clarified all these concerns. There are some minor presentation issues; the authors should make sure their final manuscript reads clearly. Some figures are hard to understand (text is small or unprocessed like "202304062000" for time stamps in Fig5); the PSD plot is also too small to see clearly to clarify concern 5. to compare against the radar resolution.

**Reviewer Concerns:**

One concern was that application is limited to a region. However, it was acknowledged that this is a limitation and may be out of scope for this paper. Other concerns were mostly addressed.

**Reviewer Scores:**

I believe all reviewers would have increased their scores to around 6/7 since their concerns were mostly addressed.

---

### Decision · Program_Chairs · 2026-01-26

Accept (Poster)